# HYPERBOLIC DEEP REINFORCEMENT LEARNING

**Edoardo Cetin**[*][†]
King's College London

**Benjamin P Chamberlain**[†]
Charm Therapeutics

**Michael M Bronstein**[†]
University of Oxford

**Jonathan J Hunt**
Twitter Inc.

## ABSTRACT

In deep reinforcement learning (RL), useful information about the state is inherently tied to its possible future successors. Consequently, encoding features that capture the hierarchical relationships between states into the model's latent representations is often conducive to recovering effective policies. In this work, we study a new class of deep RL algorithms that promote encoding such relationships by using hyperbolic space to model latent representations. However, we find that a naive application of existing methodology from the hyperbolic deep learning literature leads to fatal instabilities due to the non-stationarity and variance characterizing common gradient estimators in RL. Hence, we design a new general method that directly addresses such optimization challenges and enables stable end-to-end learning with deep hyperbolic representations. We empirically validate our framework by applying it to popular on-policy and off-policy RL algorithms on the Procgen and Atari 100K benchmarks, attaining near universal performance and generalization benefits. Given its natural fit, we hope this work will inspire future RL research to consider hyperbolic representations as a standard tool.

## 1 INTRODUCTION

Reinforcement Learning (RL) achieved notable milestones in several game-playing and robotics applications (Mnih et al., 2013; Vinyals et al., 2019; Kalashnikov et al., 2018; OpenAI et al., 2019; Lee et al., 2021). However, all these recent advances relied on large amounts of data and domain-specific practices, restricting their applicability in many important real-world contexts (Dulac-Arnold et al., 2019). We argue that these challenges are symptomatic of current deep RL models lacking a *proper prior* to efficiently learn *generalizable* features for control (Kirk et al., 2021). We propose to tackle this issue by introducing *hyperbolic geometry* to RL, as a new inductive bias for representation learning.

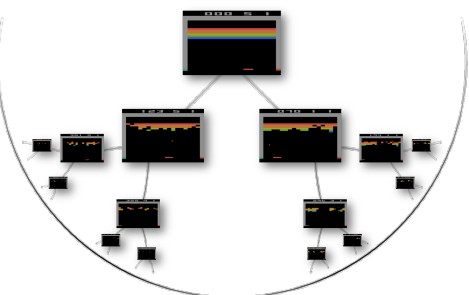

Figure 1: Hierarchical relationship between states in *breakout*, visualized in hyperbolic space.

The evolution of the state in a Markov decision process can be conceptualized as a tree, with the policy and dynamics determining the possible branches. Analogously, the same hierarchical evolution often applies to the most significant features required for decision-making (e.g., presence of bricks, location of paddle/ball in Fig. 1). These relationships tend to hold beyond individual trajectories, making hierarchy a natural basis to encode information for RL (Flet-Berliac, 2019). Consequently, we hypothesize that deep RL models should prioritize encoding precisely *hierarchically-structured* features to facilitate learning effective and generalizable policies. In contrast, we note that non-evolving features, such as the aesthetic properties of elements in the environment, are often linked with spurious correlations, hindering generalization to new states (Song et al., 2019). Similarly, human cognition also appears to learn representations of actions and elements of the environment by focusing on their underlying hierarchical relationship (Barker & Wright, 1955; Zhou et al., 2018).

Hyperbolic geometry (Beltrami, 1868; Cannon et al., 1997) provides a natural choice to efficiently encode hierarchically-structured features. A defining property of hyperbolic space is *exponential volume growth*, which enables the embedding of tree-like hierarchical data with low distortion using only a few dimensions (Sarkar, 2011). In contrast, the volume of Euclidean spaces only grows

---

[*]E-mail: edoardo.cetin@kcl.ac.uk; [†] work done while at Twitter Inc.

polynomially, requiring high dimensionality to precisely embed tree structures (Matoušek, 1990), potentially leading to higher complexity, more parameters, and overfitting. We analyze the properties of learned RL representations using a measure based on the *δ-hyperbolicity* (Gromov, 1987), quantifying how close an arbitrary metric space is to a hyperbolic one. In line with our intuition, we show that performance improvements of RL algorithms correlate with the increasing *hyperbolicity* of the discrete space spanned by their latent representations. This result validates the importance of appropriately encoding hierarchical information, suggesting that the inductive bias provided by employing hyperbolic representations would facilitate recovering effective solutions.

Hyperbolic geometry has recently been exploited in other areas of machine learning showing substantial performance and efficiency benefits for learning representations of hierarchical and graph data (Nickel & Kiela, 2017; Chamberlain et al., 2017). Recent contributions further extended tools from modern deep learning to work in hyperbolic space (Ganea et al., 2018; Shimizu et al., 2020), validating their effectiveness in both supervised and unsupervised learning tasks (Khrulkov et al., 2020; Nagano et al., 2019; Mathieu et al., 2019). However, most of these approaches showed clear improvements on smaller-scale problems that failed to hold when scaling to higher-dimensional data and representations. Many of these shortcomings are tied to the practical challenges of optimizing hyperbolic and Euclidean parameters end-to-end (Guo et al., 2022). In RL, We show the non-stationarity and high-variance characterizing common gradient estimators *exacerbates* these issues, making a naive incorporation of existing hyperbolic layers yield underwhelming results.

In this work, we overcome the aforementioned challenges and effectively train deep RL algorithms with latent hyperbolic representations end-to-end. In particular, we design *spectrally-regularized hyperbolic mappings* (S-RYM), a simple recipe combining scaling and spectral normalization (Miyato et al., 2018) that stabilizes the learned hyperbolic representations and enables their seamless integration with deep RL. We use S-RYM to build hyperbolic versions of both on-policy (Schulman et al., 2017) and off-policy algorithms (Hessel et al., 2018), and evaluate on both Procgen (Cobbe et al., 2020) and Atari 100K benchmarks (Bellemare et al., 2013). We show that our framework attains *near universal performance and generalization improvements* over established Euclidean baselines, making even general algorithms competitive with highly-tuned SotA baselines. We hope our work will set a new standard and be the first of many incorporating hyperbolic representations with RL. To this end, we share our implementation at `sites.google.com/view/hyperbolic-rl`.

## 2 PRELIMINARIES

In this section, we introduce the main definitions required for the remainder of the paper. We refer to App. A and (Cannon et al., 1997) for further details about RL and hyperbolic space, respectively.

### 2.1 REINFORCEMENT LEARNING

The RL problem setting is traditionally described as a Markov Decision Process (MDP), defined by the tuple $(S, A, P, p_0, r, \gamma)$. At each timestep $t$, an agent interacts with the environment, observing some state from the state space $s \in S$, executing some action from its action space $a \in A$, and receiving some reward according to its reward function $r : S \times A \mapsto \mathbb{R}$. The transition dynamics $P : S \times A \times S \mapsto \mathbb{R}$ and initial state distribution $p_0 : S \mapsto \mathbb{R}$ determine the evolution of the environment's state while the discount factor $\gamma \in [0, 1)$ quantifies the agent's preference for earlier rewards. Agent behavior in RL can be represented by a parameterized distribution function $\pi_\theta$, whose sequential interaction with the environment yields some trajectory $\tau = (s_0, a_0, s_1, a_1, ..., s_T, a_T)$. The agent's objective is to learn a policy maximizing its expected discounted sum of rewards over trajectories:

$$\arg\max_\theta \mathbb{E}_{\tau \sim \pi_\theta, P} \left[ \sum_{t=0}^{\infty} \gamma^t r(s_t, a_t) \right]. \tag{1}$$

We differentiate two main classes of RL algorithms with very different optimization procedures based on their different usage of the collected data. *On-policy* algorithms collect a new set of trajectories with the latest policy for each training iteration, discarding old data. In contrast, *off-policy* algorithms maintain a large *replay buffer* of past experiences and use it for learning useful quantities about the environment, such as world models and value functions. Two notable instances from each class are Proximal Policy Optimization (PPO) (Schulman et al., 2017) and Rainbow DQN (Hessel et al., 2018), upon which many recent advances have been built.

## 2.2 MACHINE LEARNING IN HYPERBOLIC SPACES

A *hyperbolic space* $\mathbb{H}^n$ is an $n$-dimensional Riemannian manifold with constant negative sectional curvature $-c$. Beltrami (1868) showed the equiconsistency of hyperbolic and Euclidean geometry using a model named after its re-discoverer, the *Poincaré ball model*. This model equips an $n$-dimensional open ball $\mathbb{B}^n = \{\mathbf{x} \in \mathbb{R}^n : c\|\mathbf{x}\| < 1\}$ of radius $1/\sqrt{c}$ with a conformal metric of the form $\mathbf{G}_{\mathbf{x}} = \lambda_{\mathbf{x}}^2 \mathbf{I}$, where $\lambda_{\mathbf{x}} = \frac{2}{1-c\|\mathbf{x}\|^2}$ is the *conformal factor* (we will omit the dependence on the curvature $-c$ in our definitions for notation brevity). The *geodesic* (shortest path) between two points in this metric is a circular arc perpendicular to the boundary with the length given by:

$$d(\mathbf{x}, \mathbf{y}) = \frac{1}{\sqrt{c}} \cosh^{-1}\left(1 + 2c \frac{\|\mathbf{x} - \mathbf{y}\|^2}{(1 - c\|\mathbf{x}\|^2)(1 - c\|\mathbf{y}\|^2)}\right). \tag{2}$$

From these characteristics, hyperbolic spaces can be viewed as a continuous analog of trees. In particular, the volume of a ball on $\mathbb{H}^n$ grows exponentially w.r.t. its radius. This property mirrors the exponential node growth in trees with constant branching factors. Visually, this makes geodesics between distinct points pass through some midpoint with lower magnitude, analogously to how tree geodesics between nodes (defined as the shortest path in their graph) must cross their closest shared parent (Fig. 2).

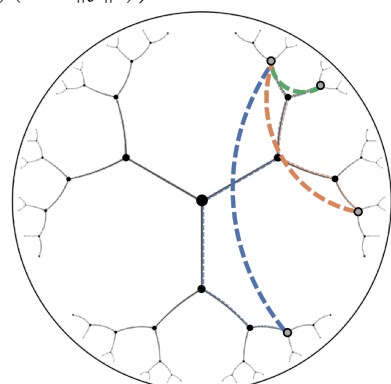

**Key operations for learning.** On a Riemannian manifold, the *exponential map* $\exp_x(\mathbf{v})$ outputs a unit step along a geodesic starting from point $x$ in the direction of an input velocity $\mathbf{v}$. It thus allows locally treating $\mathbb{H}^n$ as Euclidean space. We use the exponential map from the origin of the Poincaré ball to map Euclidean input vectors $v$ into $\mathbb{H}^n$,

Figure 2: Geodesics on $\mathbb{H}^2$ and shortest paths connecting nodes of a tree.

$$\exp_{\mathbf{0}}(\mathbf{v}) = \tanh\left(\sqrt{c}\|\mathbf{v}\|\right) \frac{\mathbf{v}}{\sqrt{c}\|\mathbf{v}\|}. \tag{3}$$

Following Ganea et al. (2018), we consider the framework of *gyrovector spaces* (Ungar, 2008) to extend common vector operations to non-Euclidean geometries, and in particular $\mathbb{H}^n$. The most basic such generalized operation is the *Mobius addition* $\oplus$ of two vectors,

$$\mathbf{x} \oplus \mathbf{y} = \frac{(1 + 2c\langle\mathbf{x}, \mathbf{y}\rangle + c\|\mathbf{y}\|^2)\mathbf{x} + (1 + c\|\mathbf{x}\|^2)\mathbf{y}}{1 + 2c\langle\mathbf{x}, \mathbf{y}\rangle + c^2\|\mathbf{x}\|^2\|\mathbf{y}\|^2}. \tag{4}$$

Next, consider a Euclidean affine transformation $f(\mathbf{x}) = \langle\mathbf{x}, \mathbf{w}\rangle + b$ used in typical neural network layers. We can rewrite this transformation as $f(\mathbf{x}) = \langle\mathbf{x} - \mathbf{p}, \mathbf{w}\rangle$ and interpret $\mathbf{w}, \mathbf{p} \in \mathbb{R}^d$ as the *normal* and *shift* parameters of a *hyperplane* $H = \{\mathbf{y} \in \mathbb{R}^d : \langle\mathbf{y} - \mathbf{p}, \mathbf{w}\rangle = 0\}$ (Lebanon & Lafferty, 2004). This allows us to further rewrite $f(\mathbf{x})$ in terms of the *signed distance* to the hyperplane $H$, effectively acting as a weighted 'decision boundary':

$$f(\mathbf{x}) = \text{sign}\left(\langle\mathbf{x} - \mathbf{p}, \mathbf{w}\rangle\right)\|\mathbf{w}\|d(\mathbf{x}, H). \tag{5}$$

This formulation allows to extend affine transformations to the Poincaré ball by considering the signed distance from a *gyroplane* in $\mathbb{B}^d$ (generalized hyperplane) $H = \{\mathbf{y} \in \mathbb{B}^d : \langle\mathbf{y} \oplus -\mathbf{p}, \mathbf{w}\rangle = 0\}$,

$$f(\mathbf{x}) = \text{sign}(\langle\mathbf{x} \oplus -\mathbf{p}\mathbf{w}\rangle)\frac{2\|\mathbf{w}\|}{\sqrt{1 - c\|\mathbf{p}\|^2}}d(\mathbf{x}, H); \ d(\mathbf{x}, H) = \frac{1}{\sqrt{c}}\sinh^{-1}\left(\frac{2\sqrt{c}|\langle\mathbf{x} \oplus -\mathbf{p}, \mathbf{w}\rangle|}{(1 - c\|\mathbf{x} \oplus -\mathbf{p}\|^2)\|\mathbf{w}\|}\right) \tag{6}$$

Similarly to recent hyperbolic deep learning work (Mathieu et al., 2019; Guo et al., 2022), we use these operations to parameterize *hybrid* neural nets: we first process the input data $\mathbf{x}$ with standard layers to produce Euclidean vectors $\mathbf{x}_E = f_E(\mathbf{x})$. Then, we obtain hyperbolic representations by applying the exponential map treating $\mathbf{x}_E$ as a velocity, $\mathbf{x}_H = \exp_{\mathbf{0}}(\mathbf{x}_E)$. Finally, we use linear operations of the form in Eq. 6 to output the set of policy and value scalars, $f_H(\mathbf{x}_H) = \{f_i(\mathbf{x}_H)\}$. We extend this model with a new stabilization (Sec 3.3) with observed benefits beyond RL (App. B).

## 3 HYPERBOLIC REPRESENTATIONS FOR REINFORCEMENT LEARNING

In this section, we base our empirical RL analysis on Procgen (Cobbe et al., 2020). This benchmark consists of 16 visual environments, with procedurally-generated random levels. Following common practice, we *train* agents using exclusively the first 200 levels of each environment and evaluate on the full distribution of levels to assess agent performance and generalization.

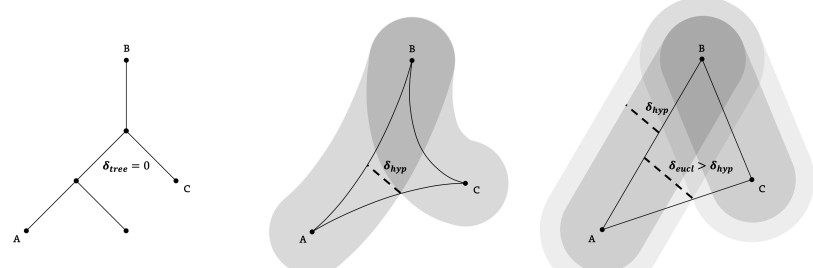

Figure 3: A geodesic space is $\delta$-hyperbolic if every triangle is $\delta$-slim, i.e., each of its sides is entirely contained within a $\delta$-sized region from the other two. We illustrate the necessary $\delta$ to satisfy this property for $\triangle ABC$ in a tree triangle (**Left**), a hyperbolic triangle (**Center**) and an Euclidean triangle (**Right**); sharing vertex coordinates. In tree triangles, $\delta_{tree} = 0$ since $\overline{AC}$ always intersects both $\overline{AB}$ and $\overline{BC}$.

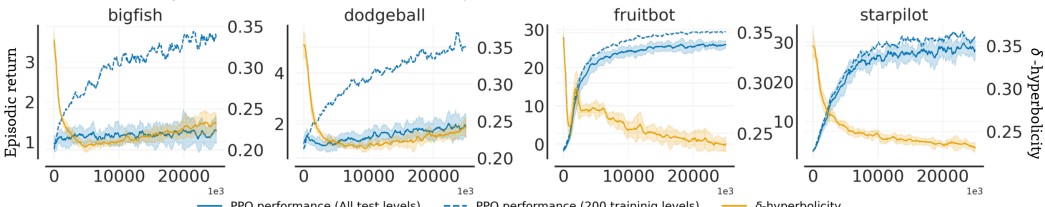

Figure 4: Performance and relative $\delta$-hyperbolicity of the final latent representations of a PPO agent.

### 3.1 THE INHERENT HYPERBOLICITY OF DEEP RL

Key quantities for each state, such as the value and the policy, are naturally related to its possible successors. In contrast, other fixed, non-hierarchical information about the environment such as its general appearance, can often be safely ignored. This divide becomes particularly relevant when considering the problem of RL generalization. For instance, Raileanu & Fergus (2021) found that agents' can overfit to spurious correlations between the value and non-hierarchical features (e.g., background color) in the observed states. Hence, we hypothesize that *effective representations* should encode features directly related to the hierarchical state relationships of MDPs.

$\delta$**-hyperbolicity.** We analyze the representation spaces learned by RL agents, testing whether they preserve and reflect this hierarchical structure. We use the $\delta$-hyperbolicity of a metric space $(X, d)$ (Gromov, 1987; Bonk & Schramm, 2011), which we formally describe in App. A.2. For our use-case, $X$ is $\delta$-hyperbolic if every possible geodesic triangle $\triangle xyz \in X$ is $\delta$-slim. This means that for every point on any side of $\triangle xyz$ there exists some point on one of the other sides whose distance is at most $\delta$. In trees, *every point belongs to at least two of its sides* yielding $\delta = 0$ (Figure 3). Thus, we can interpret $\delta$-hyperbolicity as measuring the deviation of a given metric from an exact tree metric.

The representations learned by an RL agent from encoding the collected states span some finite subset of Euclidean space $\mathbf{x}_E \in X_E \subset \mathbb{R}^n$, yielding a discrete metric space $X_E$. To test our hypothesis, we compute the $\delta$-hyperbolicity of $X_E$ and analyze how it relates to *agent performance*. Similarly to (Khrulkov et al., 2020), we compute $\delta$ using the efficient algorithm proposed by Fournier et al. (2015). To account for the scale of the representations, we normalize $\delta$ by $\text{diam}(X_E)$, yielding a *relative* hyperbolicity measure $\delta_{rel} = 2\delta/\text{diam}(X_E)$ (Borassi et al., 2015), which can span values between 0 (hyperbolic hierarchical tree-like structure) and 1 (perfectly non-hyperbolic spaces).

**Results.** We train an agent with PPO (Schulman et al., 2017) on four Procgen environments, encoding states from the latest rollouts using the representations before the final linear policy and value heads, $x_E = f_E(s)$. Hence, we estimate $\delta_{rel}$ from the space spanned by these latent encodings as training progresses. As shown in Figure 4, $\delta_{rel}$ quickly drops to low values ($0.22 - 0.28$) in the first training iterations, reflecting the largest relative improvements in agent performance. Subsequently, in the *fruitbot* and *starpilot* environments, $\delta_{rel}$ further decreases throughout training as the agent recovers high performance with a low generalization gap between the training and test distribution of levels. Instead, in *bigfish* and *dodgeball*, $\delta_{rel}$ begins to increase again after the initial drop, suggesting that the latent representation space starts losing its hierarchical structure. Correspondingly, the agent starts overfitting as test levels performance stagnates while the generalization gap with the training levels performance keeps increasing. We believe these results support our hypothesis, empirically showing the importance of encoding hierarchical features for recovering effective solu-

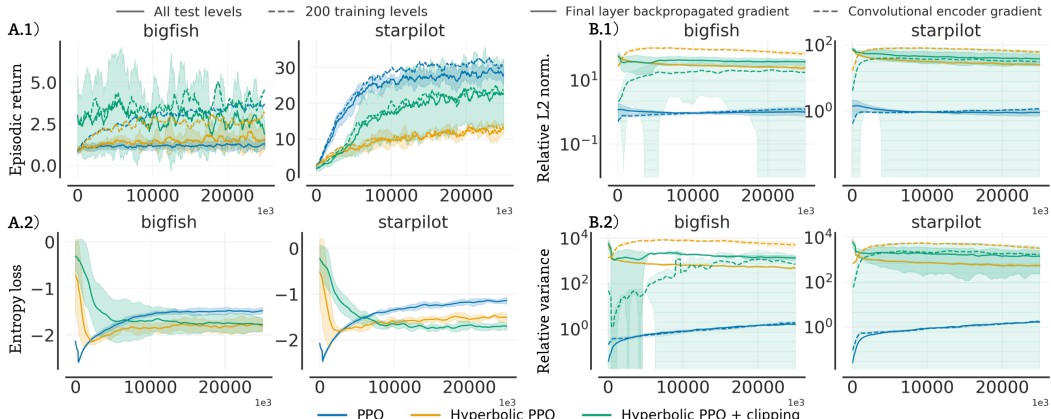

Figure 5: Analysis of key statistics for our *naive* implementations of hyperbolic PPO agents using existing practices to stabilize optimization in hyperbolic space. On the left, we display performance (**A.1**) and negative entropy (**A.2**). On the right, we display magnitudes (**B.1**) and variances (**B.2**) of the backpropagated gradients.

tions. Furthermore, they suggest that PPO's poor generalization in some environments is due to the observed tendency of Euclidean latent spaces to encode spurious features that hinder hyperbolicity.

Motivated by our findings, we propose employing hyperbolic geometry to model the latent representations of deep RL models. Representing tree-metrics in Euclidean spaces incurs non-trivial worse-case distortions, growing with the number of nodes at a rate dependent on the dimensionality (Matoušek, 1990). This property suggests that it is not possible to encode complex hierarchies in Euclidean space both efficiently and accurately, explaining why some solutions learned by PPO could not maintain their hyperbolicity throughout training. In contrast, mapping the latent representations to hyperbolic spaces of any dimensionality enables encoding features exhibiting a tree-structured relation over the data with *arbitrarily low distortion* (Sarkar, 2011). Hence, hyperbolic latent representations introduce a different *inductive bias* for modeling the policy and value function, stemming from this inherent efficiency of specifically encoding hierarchical information (Tifrea et al., 2018).

## 3.2 OPTIMIZATION CHALLENGES

**Naive integration.** We test a simple extension to PPO, mapping the latent representations of states $s \in S$ before the final linear policy and value heads $x_E = f_E(s)$ to the Poincaré ball with unitary curvature. As described in Section 2, we perform this with an exponential map to produce $x_H = \exp_0^1(x_E)$, replacing the final ReLU. To output the value and policy logits, we then finally perform a set of affine transformations in hyperbolic space, $\pi(s), V(s) = f_H(x_H) = \{f_i^1(x_H)\}_{i=0}^{|A|}$. We also consider a *clipped* version of this integration, following the recent stabilization practice from Guo et al. (2022), which entails clipping the magnitude of the latent representations to not exceed unit norm. We initialize the weights of the last two linear layers in both implementations to $100\times$ smaller values to start training with low magnitude latent representations, which facilitates the network first learning appropriate angular layouts (Nickel & Kiela, 2017; Ganea et al., 2018).

**Results.** We analyze this naive hyperbolic PPO implementation in Figure 5. As shown in part (A.1), performance is gener-

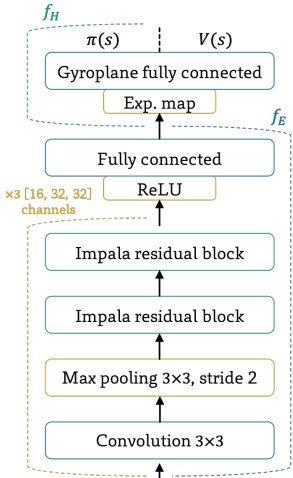

Figure 6: PPO model with an hyperbolic latent space, extending the architecture from Espeholt et al. (2018).

ally underwhelming, lagging considerably behind the performance of standard PPO. While applying the clipping strategy yields some improvements, its results are still considerably inferior on the tasks where Euclidean embeddings appear to already recover effective representations (e.g. *starpilot*). In part (A.2) we visualize the negated entropy of the different PPO agents. PPO's policy optimization objective includes both a reward maximization term, which requires an auxiliary estimator, and an entropy bonus term that can instead be differentiated exactly and optimized end-to-end. Its purpose is to push PPO agents to explore if they struggle to optimize performance with the current data.

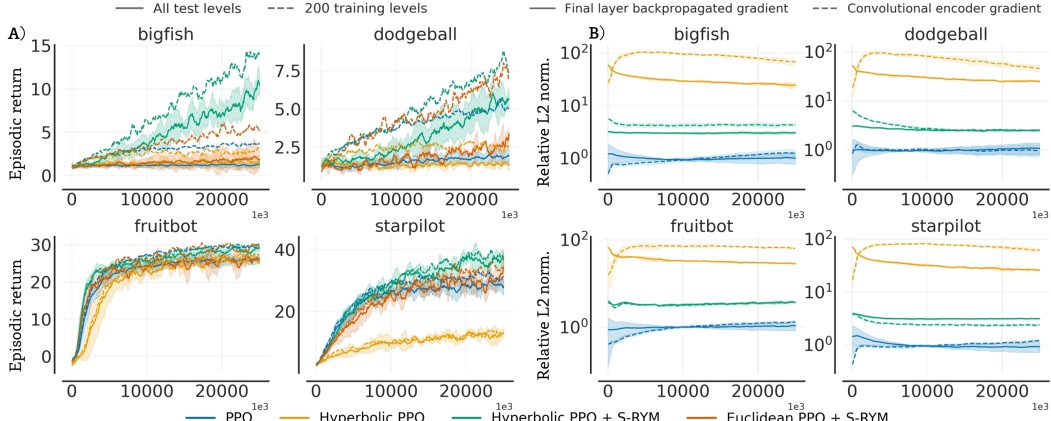

Figure 7: Analysis of hyperbolic PPO with the proposed S-RYM stabilization. We visualize performance (**A**) and gradient magnitudes (**B**) as compared to the original Euclidean and the naive hyperbolic baselines.

We note that the Hyperbolic PPO agents take significantly longer to reach higher levels of entropy in the initial training phases and are also much slower to reduce their entropy as their performance improves. These results appear to indicate the presence of optimization challenges stemming from end-to-end RL training with hyperbolic representations. Therefore, we turn our attention to analyzing the *gradients* in our hyperbolic models. In part (B.1), we visualize the magnitude of the gradients both as backpropagated from the final representations and to the convolutional encoder. In part (B.2), we also visualize the variance of the same gradients with respect to the different input states in a minibatch. We find that hyperbolic PPO suffers from a severe exploding gradients problem, with both magnitudes and variances being several orders of magnitude larger than the Euclidean baseline. Similar instabilities have been documented by much recent literature, as described in App. B. Yet, in the RL case, common stabilization techniques such as careful initialization and clipping are visibly insufficient, resulting in ineffective learning and inferior agent performance.

## 3.3 STABILIZING HYPERBOLIC REPRESENTATIONS

We hypothesize that the high variance and non-stationarity characterizing RL are the main cause of the observed optimization challenges of this naive hyperbolic PPO. implementation. Initialization and clipping have been designed for stationary ML applications with *fixed* dataset and targets. In these settings, regularizing the initial learning iterations enables the model to find appropriate angular layouts of the representations for the underlying *fixed* loss landscape. Without appropriate angular layouts, useful representations become hard to recover due to the highly non-convex spectrum of hyperbolic neural networks, resulting in failure modes with low performance (Ganea et al., 2018; López & Strube, 2020). We can intuitively see how this reliance is likely incompatible with the RL setting, where the trajectory data and loss landscape can change significantly throughout training, making early angular layouts inevitably suboptimal. We believe this is further exacerbated by the high variance gradients already characterizing policy gradient optimization (Sutton & Barto, 2018) which facilitate entering unstable learning regimes that can lead to our observed failure modes.

**Spectral norm.** Another sub-field of ML dealing with non-stationarity and brittle optimization is generative modeling with adversarial networks (GANs) (Goodfellow et al., 2014). In GAN training, the generated data and discriminator's parameters constantly evolve, making the loss landscape highly non-stationary as in the RL setting. Furthermore, the adversarial nature of the optimization makes it very brittle to exploding and vanishing gradients instabilities which lead to common failure modes (Arjovsky & Bottou, 2017; Brock et al., 2018). In this parallel literature, *spectral normalization (SN)* (Miyato et al., 2018) is a popular stabilization practice whose success made it ubiquitous in modern GAN implementations. Recent work (Lin et al., 2021) showed that a reason for its surprising effectiveness comes from *regulating* both the magnitude of the activations and their respective gradients very similarly to LeCun initialization (LeCun et al., 2012). Furthermore, when applied to the discriminator model, SN's effects appear to persist *throughout training*, while initialization strategies tend to only affect the initial iterations. In fact, they also show that ablating SN from GAN training empirically results in exploding gradients and degraded performance, closely resembling our same observed instabilities. We provide details about GANs and SN in App. A.3.

Table 1: Performance comparison for the considered versions of PPO full Procgen benchmark

| Task\Algorithm | PPO | PPO + data aug. | PPO + S-RYM | PPO + S-RYM, 32 dim. |
|---|---|---|---|---|
| **Levels distribution** | *train/test* | *train/test* | *train/test* | *train/test* |
| *bigfish* | 3.71±1 1.46±1 | 12.43±4 (+235%) 13.07±2 (+797%) | 13.27±2 (+258%) 12.20±2 (+737%) | 20.58±5 (+455%) **16.57±2 (+1037%)** |
| *bossfight* | 8.18±1 7.04±2 | 3.38±1 (-59%) 2.96±1 (-58%) | 8.61±1 (+5%) 8.14±1 (+16%) | 9.26±1 (+13%) **9.02±1 (+28%)** |
| *caveflyer* | 7.01±1 **5.86±1** | 6.08±1 (-13%) 4.89±1 (-16%) | 6.15±1 (-12%) 5.15±1 (-12%) | 6.38±1 (-9%) 5.20±1 (-11%) |
| *chaser* | 6.58±2 5.89±1 | 2.14±0 (-67%) 2.18±0 (-63%) | 6.60±2 (+0%) 7.82±1 (+33%) | 9.04±1 (+37%) **7.32±1 (+24%)** |
| *climber* | 8.66±2 5.11±1 | 7.61±1 (-12%) 5.74±2 (+12%) | 8.91±1 (+3%) 6.64±1 (+30%) | 8.32±1 (-4%) **7.28±1 (+43%)** |
| *coinrun* | 9.50±0 8.25±0 | 8.40±1 (-12%) 9.00±1 (+9%) | 9.30±1 (-2%) 8.40±0 (+2%) | 9.70±0 (+2%) **9.20±0 (+12%)** |
| *dodgeball* | 5.07±1 1.87±1 | 3.94±1 (-22%) 3.20±1 (+71%) | 7.10±1 (+40%) 6.52±1 (+248%) | 7.74±2 (+53%) **7.14±1 (+281%)** |
| *fruitbot* | 30.10±2 26.33±2 | 27.56±3 (-8%) 27.98±1 (+6%) | 30.43±1 (+1%) 27.97±3 (+6%) | 29.15±1 (-3%) **29.51±1 (+12%)** |
| *heist* | 7.42±1 2.92±1 | 4.20±1 (-43%) **3.60±0 (+23%)** | 5.40±1 (-27%) 2.70±1 (-7%) | 6.40±1 (-14%) **3.60±1 (+23%)** |
| *jumper* | 8.86±1 6.14±1 | 7.70±1 (-13%) 5.70±0 (-7%) | 9.00±1 (+2%) **6.70±1 (+9%)** | 8.50±0 (-4%) 6.10±1 (-1%) |
| *leaper* | 4.86±2 4.36±2 | 6.80±1 (+40%) 7.00±1 (+61%) | 8.00±1 (+65%) **7.30±1 (+68%)** | 7.70±1 (+59%) 7.00±1 (+61%) |
| *maze* | 9.25±0 6.50±0 | 8.50±1 (-8%) **7.10±1 (+9%)** | 9.50±0 (+3%) 6.10±1 (-6%) | 9.20±0 (-1%) **7.10±1 (+9%)** |
| *miner* | 12.95±0 9.28±1 | 9.81±0 (-24%) 9.36±2 (+1%) | 12.09±1 (-7%) 10.08±1 (+9%) | 12.94±0 (+0%) **9.86±1 (+6%)** |
| *ninja* | 7.62±1 **6.50±1** | 6.90±1 (-10%) 4.50±1 (-31%) | 6.50±1 (-15%) 6.10±1 (-6%) | 7.50±1 (-2%) 5.60±1 (-14%) |
| *plunder* | 6.92±2 6.06±3 | 5.13±0 (-26%) 4.96±1 (-18%) | 7.26±1 (+5%) 6.87±1 (+13%) | 7.35±1 (+6%) 6.68±0 (+10%) |
| *starpilot* | 30.50±5 26.57±5 | 43.43±7 (+42%) 32.41±3 (+22%) | 37.08±3 (+22%) 41.22±3 (+55%) | 41.48±4 (+36%) **38.27±5 (+44%)** |
| Average norm. score | 0.5614 0.3476 | 0.4451 (-21%) 0.3536 (+2%) | 0.5846 (+4%) 0.4490 (+29%) | **0.6326 (+13%)** **0.4730 (+36%)** |
| Median norm. score | 0.6085 0.3457 | 0.5262 (-14%) 0.3312 (-4%) | 0.6055 (+0%) 0.4832 (+40%) | **0.6527 (+7%)** **0.4705 (+36%)** |
| # Env. improvements | 0/16 0/16 | 3/16 10/16 | **11/16** 12/16 | 8/16 **13/16** |

Figure 8: Performance comparison for the considered versions of PPO agents with Euclidean and hyperbolic latent representations, increasingly lowering the number of training levels.

**S-RYM.** Inspired by these connections, we propose to counteract the optimization challenges in RL and hyperbolic representations with SN. Our implementation differs from its usual application for GANs in two main ways. First, we apply SN to all layers in the Euclidean sub-network ($f_E$), as the observed instabilities already occur in the gradients from the hyperbolic representations, but leave the final linear hyperbolic layer ($f_H$) unregularized to avoid further limiting expressivity. Second, we propose to scale the latent representations to account for their dimensionality. In particular, modeling $x_E \in \mathbb{R}^n$ by an independent Gaussian, the magnitude of the representations follows some scaled *Chi distribution* $\|x_E\| \sim \chi_n$, which we can reasonably approximate with $E[\|x_E\|] = E[\chi_n] \approx \sqrt{n}$. Therefore, we propose to rescale the output of $f_E$ by $1/\sqrt{n}$, such that modifying the dimensionality of the representations should not significantly affect their magnitude before mapping them $\mathbb{H}^n$. We call this general stabilization recipe *spectrally-regularized hyperbolic mappings* (S-RYM).

**Results.** As shown in Figure 7, integrating S-RYM with our hyperbolic RL agents appears to resolve their optimization challenges and considerably improve the Euclidean baseline's performance (A). To validate that the performance benefits are due to the hyperbolic geometry of the latent space, we also evaluate a baseline using SN and rescaling in Euclidean space, which fails to attain consistent improvements. Furthermore, S-RYM maintains low gradient magnitudes (B), confirming its effectiveness to stabilize training. In App. E.1, we also show that SN and rescaling are both crucial for S-RYM. Thus, in the next section, we evaluate hyperbolic deep RL on a large-scale, analyzing its efficacy and behavior across different benchmarks, RL algorithms, and training conditions.

## 4 EXTENSIONS AND EVALUATION

To test the generality of our hyperbolic deep RL framework, in addition to the on-policy PPO we also integrate it with the off-policy Rainbow DQN algorithm (Hessel et al., 2018). Our implementations use the same parameters and models specified in prior traditional RL literature, without any additional tuning. Furthermore, in addition to the full Procgen benchmark (16 envs.) we also evaluate on

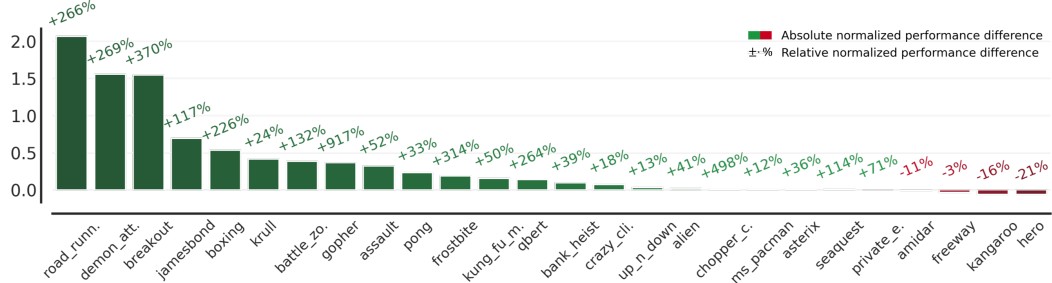

Figure 9: Absolute difference in normalized performance (**Y-axis**) and relative improvements (**Above bars**) from integrating hyperbolic representations with S-RYM onto our Rainbow implementation.

the popular Atari 100K benchmark (Bellemare et al., 2013; Kaiser et al., 2020) (26 envs.), repeating for 5 random seeds. We provide all details about benchmarks and implementations in App. C.

**Generalization on Procgen.** Given the documented representation efficiency of hyperbolic space, we evaluate our hyperbolic PPO implementation also reducing the dimensionality of the final representation to 32 (see App. E.2), with relative compute and parameter efficiency benefits. We compare our regularized hyperbolic PPO with using data augmentations, a more traditional way of encoding inductive biases from inducing invariances. We consider random crop augmentations from their popularity and success in modern RL. As shown in Table 1, our hyperbolic PPO implementation with S-RYM appears to yield conspicuous performance gains on most of the environments. At the same time, reducing the size of the representations provides even further benefits with significant improvements in 13/16 tasks. In contrast, applying data augmentations yields much lower and inconsistent gains, even hurting on some tasks where hyperbolic RL provides considerable improvements (e.g. *bossfight*). We also find that test performance gains do not always correlate with gains on the specific 200 training levels, yielding a significantly reduced generalization gap for the hyperbolic agents. We perform the same experiment but apply our hyperbolic deep RL framework to Rainbow DQN with similar results, also obtaining significant gains in 13/16 tasks, as reported in App. D.1.

We also evaluate the robustness of our PPO agents to encoding *spurious* features, only relevant for the training levels. In particular, we examine tasks where PPO tends to perform well and consider lowering the training levels from 200 to 100, 50, and 25. As shown in Figure 8, the performance of PPO visibly drops at each step halving the number of training levels, suggesting that the Euclidean representations overfit and lose their original efficacy. In contrast, hyperbolic PPO appears much more robust, still surpassing the original PPO results with only 100 training levels in *fruitbot* and 50 in *starpilot*. While also applying data augmentation attenuates the performance drops, its effects appear more limited and inconsistent, providing almost null improvements for *starpilot*.

**Sample-efficiency on Atari 100K.** We focus on the performance of our hyperbolic Rainbow DQN implementation, as the severe data limitations of this benchmark make PPO and other on-policy algorithms impractical. We show the absolute and relative per-environment performance changes from our hyperbolic RL framework in Figure 9, and provide

Table 2: Aggregate results on Atari 100K

| Metric\Algorithm | Rainbow | Rainbow + S-RYM |
|---|---|---|
| Human norm. mean | 0.353 | **0.686 (+93%)** |
| Human norm. median | 0.259 | **0.366 (+41%)** |
| Super human scores | 2 | **5** |

aggregate statistics in Table 2. Also on this benchmark, the exact same hyperbolic deep RL framework provides consistent and significant benefits. In particular, we record improvements on 22/26 Atari environments over the Euclidean baseline, almost doubling the final human normalized score.

**Considerations and comparisons.** Our results empirically validate that introducing hyperbolic representations to shape the prior of deep RL models is both remarkably general and effective. We record almost universal improvements on two fundamentally different RL algorithms, considering both generalizations to new levels from millions of frames (Procgen) and to new experiences from only 2hrs of total play time (Atari 100K). Furthermore, our hyperbolic RL agents outperform the scores reported in most other recent advances, coming very close to the current SotA algorithms which incorporate different expensive and domain-specialized auxiliary practices (see App. D.2-D.3). Our approach is also orthogonal to many of these advances and appears to provide compatible and complementary benefits (see App. E.3). Taken together, we believe these factors show the great potential of our hyperbolic framework to become a standard way of parameterizing deep RL models.

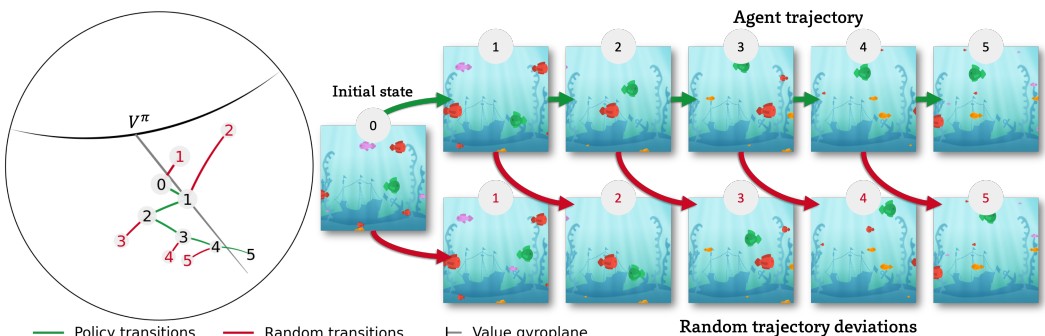

Figure 10: Visualization of 2-dimensional hyperbolic embeddings in the *bigfish* environment as we progress through a trajectory, encoding states from either policy transitions or random transitions (details in App. D.4).

**Representations interpretation.** We train our hyperbolic PPO agent with only *2-dimensional* representations, which still remarkably provide concrete generalization benefits over Euclidean PPO (App. D.4). Then, we analyze how these representations evolve within trajectories, mapping them on the Poincaré disk and visualizing the corresponding states. We observe a recurring *cyclical* behavior, where the magnitude of the representations monotonically increases within subsets of the trajectory as more obstacles/enemies appear. We show this in Fig. 10 and Fig. 12, comparing the representations of on-policy states sampled at constant intervals with trajectory deviations from executing random behavior. We observe the representations form tree-like structures, with the magnitudes in the on-policy states growing in the direction of the Value function's *gyroplane*'s normal. This intuitively reflects that as new elements appear the agent recognizes a larger opportunity for rewards, yet, requiring a finer level of control as distances to the policy gyroplanes will also grow exponentially, reducing entropy. Instead, following random deviations, magnitudes grow in directions orthogonal to the Value gyroplane's normal. This still reflects the higher precision required for optimal decision-making, but also the higher uncertainty to obtain future rewards from worse states.

## 5 RELATED WORK

Generalization is a key open problem in RL (Kirk et al., 2021). End-to-end training of deep models with RL objectives has been shown prone to overfitting from spurious features only relevant in the observed transitions (Song et al., 2019; Bertran et al., 2020). To address this, prior work considered different data augmentation strategies (Laskin et al., 2020b; Yarats et al., 2021a; Cobbe et al., 2019), and online adaption methods on top to alleviate engineering burdens (Zhang & Guo, 2021; Raileanu et al., 2020). Alternative approaches have been considering problem-specific properties of the environment (Zhang et al., 2020; Raileanu & Fergus, 2021), auxiliary losses (Laskin et al., 2020a; Schwarzer et al., 2020), and frozen pre-trained layers (Yarats et al., 2021b; Stooke et al., 2021). Instead, we propose to encode a new inductive bias making use of the geometric properties of hyperbolic space, something orthogonal and likely compatible with most such prior methods.

While hyperbolic representations found recent popularity in machine learning, there have not been notable extensions for deep RL (Peng et al., 2021). Most relatedly, Tiwari & Prannoy (2018) proposed to produce hyperbolic embeddings of the state space of tabular MDPs to recover options (Sutton et al., 1999). Yet, they did not use RL for learning, but fixed data and a supervised loss based on the co-occurrence of states, similarly to the original method by Nickel & Kiela (2017).

## 6 DISCUSSION AND FUTURE WORK

In this work, we introduce hyperbolic geometry to deep RL. We analyze training agents using latent hyperbolic representations and propose *spectrally-regularized hyperbolic mappings*, a new stabilization strategy that overcomes the observed optimization instabilities. Hence, we apply our framework to obtain hyperbolic versions of established on-policy and off-policy RL algorithms, which we show substantially outperform their Euclidean counterparts in two popular benchmarks. We provide numerous results validating that hyperbolic representations provide deep models with a more suitable prior for control, with considerable benefits for generalization and sample-efficiency. We share our implementation to facilitate future RL advances considering hyperbolic space as a new, general tool.

## ETHICS STATEMENT

We proposed to provide deep RL models with a more suitable prior for learning policies, using hyperbolic geometry. In terms of carbon footprint, our implementation does not introduce additional compute costs for training, and even appears to perform best with more compact representation sizes. Consequently, given the nature of our contribution, its ethical implications are bound to the implications of advancing the RL field. In this regard, as autonomous agents become more applicable, poor regulation and misuse may cause harm. Yet, we believe these concerns are currently out-weighted by the field's significant positive potential to advance human flourishing.

## REPRODUCIBILITY STATEMENT

We provide detailed descriptions of our integration of hyperbolic space, experimental setups, and network architectures in Section 3 and also Appendix B. We provide all details, including a full list of hyper-parameters, in Appendix C. We currently shared an anonymized version of our code to reproduce the main experiments in the supplementary material. We shared our open-source implementation to facilitate future extensions.

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

APPENDIX

# A  EXTENDED BACKGROUND

## A.1  RL ALGORITHMS DESCRIPTIONS

Continuing from section 2.1, we provide an overview of standard RL definitions and the deep RL algorithms we use in this work.

Two important functions in RL are the value function and the action-value function (also called the $Q$ function). These quantify, for policy $\pi$, the expected sum of discounted future rewards given any initial fixed state or state-action pair, respectively:

$$V^\pi(s_t) = \mathbb{E}_{a_t, s_{t+1}, a_{t+1}, \cdots \sim \pi_\theta, P} \left[ \sum_{t'=0}^\infty \gamma^{t'} r(s_{t+t'}, a_{t+t'}) \right],$$
$$Q^\pi(s_t, a_t) = r(s_t, a_t) + \gamma \mathbb{E}_{s_{t+1} \sim P} \left[ V^\pi(s_{t+1}) \right]. \tag{7}$$

Relatedly, the advantage function $A^\pi(s_t, a_t) = Q^\pi(s_t, a_t) - V^\pi(s_t)$ quantifies the expected improvement from executing any given action $a_t$ from $s_t$ rather than following the policy. These functions summarize the future evolution of an MDP and are often parameterized and learned auxiliary to or even in-place of the policy model.

**On-policy methods.** Modern on-policy RL algorithms collect a new set of trajectories at each iteration with the current policy, discarding old data. They use these trajectories to learn the current policy's value function and recover a corresponding advantage function from the observed Monte-Carlo returns, using techniques such as the popular Generalized Advantage Estimator (GAE) (Schulman et al., 2015). The estimated advantages $A^{GAE}$ are then used to compute the *policy gradient* and update the policy, maximizing the probability of performing the best-observed actions (Sutton & Barto, 2018). Since the values of $A^{GAE}$ are based on a limited set of trajectories, on-policy methods generally suffer from high-variance targets and gradients (Pendrith et al., 1997; Mannor et al., 2007; Wu et al., 2018). Proximal Policy Optimization (PPO) (Schulman et al., 2017) is one of the most established on-policy algorithms that attenuates these issues by taking conservative updates, restricting the policy update from making larger than $\epsilon$ changes to the probability of executing any individual action. PPO considers the ratio between the updated and old policy probabilities $R_\pi(a_t|s_t) = \frac{\pi_\theta(a_t|s_t)}{\pi_{old}(a_t|s_t)}$ to optimize a pessimistic clipped objective of the form:

$$\min\{R_\pi(a_t|s_t)A^{GAE}(s_t, a_t), \text{clip}(R_\pi(a_t|s_t), 1 - \epsilon, 1 + \epsilon)A^{GAE}(s_t, a_t)\}. \tag{8}$$

As mentioned in the main text, PPO also includes a small entropy bonus to incentivize exploration and improve data diversity. This term can be differentiated and optimized without any estimator since we have full access to the policy model and its output logits, independently of the collected data.

**Off-policy methods.** In contrast, off-policy algorithms generally follow a significantly different optimization approach. They store many different trajectories collected with a mixture of old policies in a large *replay buffer*, $B$. They use this data to directly learn the $Q$ function for the optimal greedy policy with a squared loss based on the Bellman backup (Bellman, 1957):

$$E_{(s_t, a_t, s_{t+1}, r_t) \in B} \left[ Q(s_t, a_t) - \left( r_t + \max_{a'} Q(s_{t+1}, a') \right) \right]^2 \tag{9}$$

(Bellman, 1957). Agent behavior is then implicitly defined by the *epsilon-greedy* policy based on the actions with the highest estimated Q values. We refer to the deep Q-networks paper (Mnih et al., 2013) for a detailed description of the seminal DQN algorithm. Rainbow DQN (Hessel et al., 2018) is a modern popular extension that introduces several auxiliary practices from proposed orthogonal improvements, which they show provide compatible benefits. In particular, they use n-step returns (Sutton & Barto, 2018), prioritized experience replay (Schaul et al., 2016), double Q-learning (Hasselt, 2010), distributional RL (Bellemare et al., 2017), noisy layers (Fortunato et al., 2018), and a dueling network architecture (Wang et al., 2016).

## A.2 $\delta$-HYPERBOLICITY

$\delta$-hyperbolicity was introduced by Gromov (1987) as a criterion to quantify how hyperbolic a metric space $(X, d)$ is. We can express $\delta$-hyperbolicity in terms of the *Gromov product*, defined for $x, y \in X$ at some base point $r \in X$ as measuring the defect from the triangle inequality:

$$(x|y)_r = \frac{1}{2}(d(x, r) + d(r, y) - d(x, y)). \tag{10}$$

Then, $X$ is $\delta$-*hyperbolic* if for all base points $r \in X$ and for any three points $x, y, z \in X$ the Gromov product between $x$ and $y$ is lower than the minimum Gromov product of the other pairs by at most some slack variable $\delta$:

$$(x|y)_r \geq \min((x|y)_r, (x|y)_r) - \delta. \tag{11}$$

In our case (a complete finite-dimensional path-connected Riemannian manifold, which is a geodesic metric space), $\delta$-hyperbolicity means that for every point on one of the sides of a geodesic triangle $\triangle xyz$, there exists some other point on one of the other sides whose distance is at most $\delta$, or in other words, geodesic triangles are $\delta$-slim. In trees, the three sides of a triangle must all intersect at some midpoint (Figure 3). Thus, *every point belongs to at least two of its sides* yielding $\delta = 0$. Thus the $\delta$-hyperbolicity can be interpreted as measuring the deviation of a given metric from an exact tree metric.

## A.3 GENERATIVE ADVERSARIAL NETWORKS AND SPECTRAL NORMALIZATION

**GANs.** In GAN training, the goal is to obtain a generator network to output samples resembling some 'true' target distribution. To achieve this, Goodfellow et al. (2014) proposed to alternate training of the generator with training an additional discriminator network, tasked to distinguish between the generated and true samples. The generator's objective is then to maximize the probability of its own samples according to the current discriminator, backpropagating directly through the discriminator's network. Since both the generated data and discriminator's network parameters constantly change from this alternating optimization, the loss landscape of GANs is also highly non-stationary, resembling, to some degree, the RL setting. As analyzed by several works, the adversarial nature of the optimization makes it very brittle to exploding and vanishing gradients instabilities (Arjovsky & Bottou, 2017; Brock et al., 2018) which often result in common failure modes from severe divergence or stalled learning (Lin et al., 2021). Consequently, numerous practices in the GAN literature have been proposed to stabilize training (Radford et al., 2015; Arjovsky et al., 2017; Gulrajani et al., 2017b). Inspired by recent work, in this work, we focus specifically on spectral normalization (Miyato et al., 2018), one such practice whose recent success made in ubiquitous in modern GAN implementations.

**Spectral normalization.** In the adversarial interplay characterizing GAN training, instabilities commonly derive from the gradients of the discriminator network, $f_D$ (Salimans et al., 2016). Hence, Miyato et al. (2018) proposed to regularize the spectral norm of discriminator's layers, $l_j \in f_D$, i.e., the largest singular values of the weight matrices $\|W_j^{WN}\|_{sn} = \sigma(W_j^{SN})$, to be approximately one. Consequently, spectral normalization effectively bounds the Lipschitz constant of the whole discriminator network since, $\|f_D\|_{Lip} \leq \prod_{j=1}^{L} \|l_j\|_{Lip} \leq \prod_{j=1}^{L} \|W_j^{SN}\|_{sn} = 1$. In practice, the proposed implementation approximates the largest singular value of some original unconstrained weight matrices by running power iteration (Golub & Van der Vorst, 2000). Thus, it recovers the spectrally-normalized weights with a simple re-parameterization, dividing the unconstrained weights by their relative singular values $W_j^{SN} = \frac{W_j}{\sigma(W_j)}$. As mentioned in the main text, recent work (Lin et al., 2021) showed that one of the main reasons for the surprising effectiveness of spectral normalization in GAN training comes from *effectively regulating* both the magnitude of the activations and their respective gradients, very similarly to LeCun initialization (LeCun et al., 2012). Furthermore, when applied to the discriminator, spectral normalization's effects appear to persist *throughout training*, while initialization strategies tend to only affect the initial iterations. In fact, in Figure 2 of their paper, they also show that ablating spectral normalization empirically results in exploding gradients and degraded performance, closely resembling our same observed instabilities in Figure 5 (B).

S-RYM entails applying spectral normalization to all the layers in the Euclidean encoder sub-network, as already after backpropagation through the final latent hyperbolic representation we observe exploding high-variance gradients. We leave the final linear transformation in hyperbolic space

unregularized as we do not want unnecessarily restrict the expressivity of the model. Furthermore, there is also no direct way of performing power iteration with our final layer parameterization (see Section 2.2). We note that by applying spectral normalization to all layers, we would enforce our models of the value and policy to be 1-Lipschitz, a property that is likely not reflective of the true optimal policy and value functions. We validate this hypothesis in Appendix E.4 by enforcing Lipschitz continuity with gradient penalties (Gulrajani et al., 2017b) on top of our regularized Hyperbolic PPO implementation. For Euclidean PPO, preliminary experiments confirmed that enforcing the model to be 1-Lipschitz by applying either spectral normalization to all layers or gradient penalties leads to worse results than applying S-RYM. This intuition is also consistent with other recent works that studied the application of spectral normalization for reinforcement learning (Bjorck et al., 2021; Gogianu et al., 2021). Yet, these works also observed performance benefits when applying spectral normalization exclusively to particular layers of the model. These empirical insights could inform future improvements for S-RYM to retain the stability benefits of spectral normalization with even less restrictive regularization.

## B  STABILIZATION OF HYPERBOLIC REPRESENTATIONS

One of the main challenges of incorporating hyperbolic geometry with neural networks comes from end-to-end optimization of latent representations and parameters located in hyperbolic space. For instance, numerical issues and vanishing gradients occur as representations get too close to either the origin or the boundary of the Poincaré ball (Ganea et al., 2018). Moreover, training dynamics can tend to push representations towards the boundary, slowing down learning and make optimization problems of earlier layers ineffective (Guo et al., 2022). A number of methods have been used to help stabilize learning of hyperbolic representations including constraining the representations to have a low magnitude early in training, applying clipping and perturbations (Ganea et al., 2018; Khrulkov et al., 2020), actively masking invalid gradients (Mathieu et al., 2019), and designing initial 'burn-in' periods of training with lower learning rates (Nickel & Kiela, 2017; Bécigneul & Ganea, 2019). More recently Guo et al. (2022) also showed that very significant magnitude clipping of the latent representations can effectively attenuate these numerical and learning instabilities when training hyperbolic classifiers for popular image classification benchmarks.

### B.1  MAGNITUDE CLIPPING

Guo et al. (2022) recently proposed to apply significant clipping of the magnitude of the latent representations when using hyperbolic representations within deep neural networks. As in our work, they also consider a *hybrid* architecture where they apply an exponential map before the final layer to obtain latent representations in hyperbolic space. They apply the proposed clipping to constrain the input vector of the exponential map to not exceed unit norm, producing hyperbolic representations via:

$$x_H = \exp_0^1 \left( \min \left\{ 1, \frac{1}{||x_E||} \right\} \times x_E \right). \tag{12}$$

The main motivation for this practice is to constrain representation magnitudes, which the authors linked to a vanishing gradient phenomenon when training on standard image classification datasets (Krizhevsky et al., 2009; Deng et al., 2009). However, a side effect of this formulation is that the learning signal from the representations exceeding a magnitude of 1 will solely convey information about the representation's direction and not its magnitude. Since the authors do not share their implementation, we tested applying their technique as described in the paper. We found some benefits in additionally initializing the parameters of the last two linear layers (in Euclidean and hyperbolic space) to $100\times$ smaller values to facilitate learning initial angular layouts.

### B.2  IMAGE CLASSIFICATION EXPERIMENTS

To empirically validate and analyze our clipping implementation we consider evaluating deep hyperbolic representations on image classification tasks, following the same training practices and datasets from Guo et al. (2022). In particular, we utilize a standard ResNet18 architecture (He et al., 2016) and test our network on CIFAR10 and CIFAR100 (Krizhevsky et al., 2009). We optimize the Euclidean parameters of the classifier using stochastic gradient descent with momentum and the hyperbolic parameters using its Riemmanian analogue (Bonnabel, 2013). We train for 100 epochs with

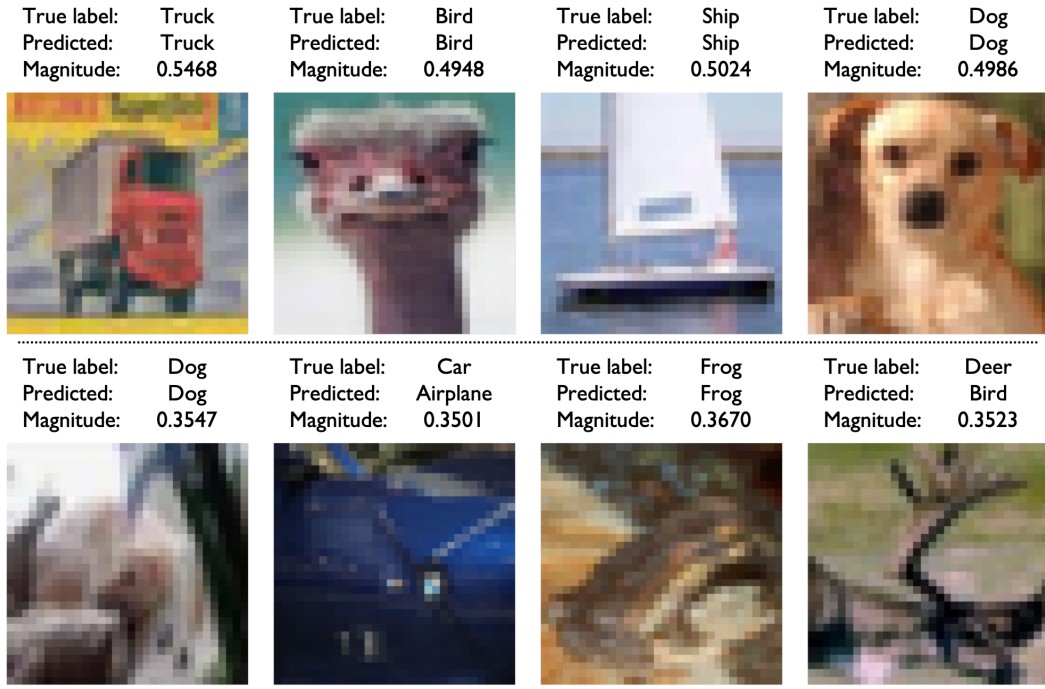

Figure 11: Visualization of test images from CIFAR10, with the corresponding final latent representations magnitudes from our hyperbolic ResNet18 classifier implemented with S-RYM. We sample datapoints with the 5% highest magnitudes (**Top**) and the 5% lowest magnitudes (**Bottom**).

an initial learning rate of 0.1 and a cosine schedule (Loshchilov & Hutter, 2017), using a standard batch size of 128. We repeat each experiment 3 times, recording the final top-1 classification accuracy together with the latent representations in Euclidean space right before applying the exponential map at the final layer.

Table 3: Performance results on standard image classification benchmarks

| **CIFAR10 with ResNet18** | | | |
|---|---|---|---|
| **Metric\Architecture** | Euclidean | Hyperbolic + Clipping | Hyperbolic + S-RYM |
| Top-1 accuracy | $94.92 \pm 0.19$ | $94.81 \pm 0.17$ | $\mathbf{95.12 \pm 0.09}$ |
| L2 representations magnitudes | 5.846 | 1.00 | 0.481 |
| Magnitudes standard deviation | 0.747 | 0.00 | 0.039 |
| **CIFAR100 with ResNet18** | | | |
| **Metric\Architecture** | Euclidean ResNet18 | Hyperbolic + Clipping | Hyperbolic + S-RYM |
| Top-1 accuracy | $76.86 \pm 0.23$ | $76.75 \pm 0.23$ | $\mathbf{77.49 \pm 0.35}$ |
| L2 representations magnitudes | 11.30 | 1.00 | 0.852 |
| Magnitudes standard deviation | 1.571 | 0.00 | 0.076 |

In Table 3, we report the different classifiers' performance together with the mean and standard deviation of the representations' magnitudes from the images in the test set. The performance of the clipped hyperbolic classifier is very close to the performance of the Euclidean classifier, matching Guo et al. (2022)'s results and validating our implementation. However, the learned representations' magnitudes soon overshoot the clipping threshold and get mapped to constant-magnitude vectors throughout training. Therefore, the model will effectively stop optimizing for the representations' magnitudes and only focus on their *unit direction*. As volume and distances on the Poincaré ball grow expoentially with radius, the magnitude component of the hyperbolic representations is pre-

cisely what facilitates encoding hierarchical information, providing its intuitive connection with tree structures. Hence, the resulting clipped 'hyperbolic' space spanned by the clipped latent representations will lose its *defining* degree of freedom and approximately resemble an $n-1$-dimensional Euclidean space with a rescaled metric, potentially explaining its performance similarity with standard Euclidean classifiers. Even though the focus of our work is not image classification, we find S-RYM's performance remarkably recovers and even marginally exceeds the performance of both the Euclidean and the clipped hyperbolic classifiers on these saturated benchmarks. Furthermore, its representations do not explode and maintain magnitude diversity, enabling to more efficiently capture the relative hierarchical nature of image-classification benchmarks (Khrulkov et al., 2020). Overall, these results suggest that clipping simply treats the *symptoms* of the instabilities caused by end-to-end large scale training by essentially resorting back to Euclidean representations for image classification.

Analyzing the magnitude component of the latent representations for our hyperbolic classifier with S-RYM, we find it correlates with classification performance. For instance, on CIFAR10 the test performance on the images with representations's with the top 5% magnitudes is 97.17%, while for the bottom 5% is 79.64%. Furthermore, we display some samples from these two distinct groups in Figure 11. From these results and visualizations, it appears that the hyperbolic hierarchical structure serves to encode the degree of uncertainty to disambiguate between multiple image labels due to the blurriness and varying difficulty of the CIFAR10 datapoints. Hence, we believe the observed accuracy improvements of our hyperbolic classifier might be specifically due to more efficiently capturing this specific hierarchical property of the considered datasets.

## C  IMPLEMENTATION AND EXPERIMENT DETAILS

We provide details of the experimental settings and implementations with the corresponding hyper-parameters for both our Proximal Policy Optimization (PPO) (Schulman et al., 2017) and Rainbow DQN experiments (Hessel et al., 2018). We consider these two main baselines since they are two of the most studied algorithms in the recent RL literature onto which many other recent advances also build upon (e.g., (Cobbe et al., 2021; Laskin et al., 2020b; Mohanty et al., 2021; Raileanu et al., 2020; Raileanu & Fergus, 2021; Yarats et al., 2021a; Van Hasselt et al., 2019; Laskin et al., 2020a; Schwarzer et al., 2020)). Furthermore, PPO and Rainbow DQN are based on the main families of model-free RL algorithms, with very distinct properties as described in Appendix A.1. Hence, unlike most prior advances, we do not constrain our analysis to a single class of methods, empirically showing the generality of hyperbolic deep RL. Our implementations closely follow the reported details from recent research, and were not tuned to facilitate our integration of hyperbolic representations. The main reason for this choice is that we wanted to avoid introducing additional confounding factors from our evaluation of hyperbolic representations, as ad-hoc tuning frequently plays a significant role in RL performance (Islam et al., 2017).

We implemented the proposed dimensionality-based rescaling, followed by the exponential map and the linear layer in hyperbolic space as a single Pytorch module (the PoincarePlaneDistance class in the shared code). Our implementation allows us to easily integrate our framework with existing neural network models by simply swapping the final layer with our new module. We would like to acknowledge the *Geoopt* optimization library (Kochurov et al., 2020), which we used to efficiently train and store the network parameters located in hyperbolic space. We also make use of the Hydra library (Yadan, 2019) to facilitate storing hyper-parameters and quickly specifying the different algorithm variations for the experiments in our work.

### C.1  BENCHMARKS

**Procgen.** The Procgen generalization benchmark (Cobbe et al., 2020) consists of 16 game environments with procedurally-generated random levels. The state spaces of these environments consist of the RGB values from the 64x64 rescaled visual renderings. Following common practice and the recommended settings, we consider training agents using exclusively the first 200 levels of each environment and evaluate on the full distribution of levels to assess agent performance and generalization. Furthermore, we train for 25M total environment steps and record final training/test performance collected across the last 100K steps averaged over 100 evaluation rollouts.

**Atari 100K.** The Atari 100K benchmark (Kaiser et al., 2020) is based on the seminal problems from the Atari Learning Environment (Bellemare et al., 2013). In particular, this benchmark consists of 26 different environments and only 100K total environment steps for learning each, corresponding roughly to 2hrs of play time. The environments are modified with the specifications from (Machado et al., 2018), making the state spaces of these environments 84x84 rescaled visual renderings and introducing randomness through *sticky actions*. We note that this is a significantly different setting than Procgen, testing the bounds for the sample efficiency of RL agents.

## C.2  PPO IMPLEMENTATION

Our PPO implementation follows the original Procgen paper (Cobbe et al., 2020), which entails a residual convolutional network (Espeholt et al., 2018) and produces a final 256-dimensional latent representation with a shared backbone for both the policy and value function. Many prior improvements over PPO for on-policy learning have been characterized by either introducing auxiliary domain-specific practices, increasing the total number of parameters, or performing additional optimization phases - leading to significant computational overheads (Cobbe et al., 2021; Raileanu & Fergus, 2021; Mohanty et al., 2021). Instead, our approach strives for an orthogonal direction by simply utilizing hyperbolic geometry to facilitate encoding hierarchically-structured features into the final latent representations. Thus, it can be interpreted as a new way to *modify the inductive bias of deep learning models for reinforcement learning*.

Table 4: PPO hyper-parameters used for the Procgen generalization benchmark

| PPO hyperparameters | |
|---|---|
| Parallel environments | 64 |
| Stacked input frames | 1 |
| Steps per rollout | 16384 |
| Training epochs per rollout | 3 |
| Batch size | 2048 |
| Normalize rewards | True |
| Discount $\gamma$ | 0.999 |
| GAE $\lambda$ (Schulman et al., 2015) | 0.95 |
| PPO clipping | 0.2 |
| Entropy coefficient | 0.01 |
| Value coefficient | 0.5 |
| Shared network | True |
| Impala stack filter sizes | 16, 32, 32 |
| Default latent representation size | 256 |
| Optimizer | *Adam* (Kingma & Ba, 2015) |
| Optimizer learning rate | $5 \times 10^{-4}$ |
| Optimizer stabilization constant ($\epsilon$) | $1 \times 10^{-5}$ |
| Maximum gradient norm. | 0.5 |

In Table 4 we provide further details of our PPO hyper-parameters, as also described by the original Procgen paper (Cobbe et al., 2020). When using hyperbolic latent representations, we optimize the hyperbolic weights of the final Gyroplane linear layer with the Riemannian Adam optimizer (Bécigneul & Ganea, 2019), keeping the same learning rate and other default parameters. As per common practice in on-policy methods, we initialize the parameters of the final layer with $100\times$ times lower magnitude. We implemented the naive hyperbolic reinforcement learning implementations introduced in Subsection 3.2 by initializing also the weights of the preceding layer with $100\times$ lower magnitudes to facilitate learning appropriate angular layouts in the early training iterations. We found our S-RYM stabilization procedure also enable to safely remove this practice with no effects on performance.

## C.3 RAINBOW IMPLEMENTATIONS

Our implementation of Rainbow DQN uses the same residual network architecture as our PPO implementation (Espeholt et al., 2018) but employs a final latent dimensionality of 512, as again specified by Cobbe et al. (2020). Since Cobbe et al. (2020) do not open-source their Rainbow implementation and do not provide many of the relative details, we strive for a simple implementation removing unnecessary complexity and boosting overall efficiency. Following Castro et al. (2018), we only consider Rainbow DQN's three most significant advances over vanilla DQN (Mnih et al., 2013): distributional critics (Bellemare et al., 2017), prioritized experience replay (Schaul et al., 2016), and n-step returns (Sutton & Barto, 2018). While the methodology underlying off-policy algorithms is fundamentally different from their on-policy counterparts, we apply the same exact recipe of integrating hyperbolic representations in the final layer, and compare against the same variations and baselines.

Table 5: Rainbow DQN hyper-parameters used for the Procgen generalization benchmark

| Rainbow DQN Procgen hyperparameters | |
|---|---|
| Parallel environments | 64 |
| Stacked input frames | 1 |
| Replay buffer size | 1.28M |
| Batch size | 512 |
| Minimum data before training | 32K steps |
| Update network every | 256 env. steps |
| Update target network every | 12800 env. steps |
| $\epsilon$-greedy exploration schedule | 1→0.01 in 512K steps |
| Discount $\gamma$ | 0.99 |
| N-step | 3 |
| Use dueling (Wang et al., 2016) | False |
| Use noisy layers (Fortunato et al., 2018) | False |
| Use prioritized replay (Schaul et al., 2016) | True |
| Use distributional value (Bellemare et al., 2017) | True |
| Distributional bins | 51 |
| Maximum distributional value | 10 |
| Minimum distributional value | -10 |
| Impala stack filter sizes | 16, 32, 32 |
| Default latent representation size | 512 |
| Optimizer | *Adam* (Kingma & Ba, 2015) |
| Optimizer learning rate | $5 \times 10^{-4}$ |
| Optimizer stabilization constant ($\epsilon$) | $1 \times 10^{-5}$ |
| Maximum gradient norm. | 0.5 |

In Table 5 we provide details of our Rainbow DQN hyper-parameters. We note that sampling of off-policy transitions with n-step returns requires retrieving the future $n$ rewards and observations. To perform this efficiently while gathering multiple transitions from the parallelized environment, we implemented a parellalized version of a segment tree. In particular, this extends the original implementation proposed by Schaul et al. (2016), through updating a set of segment trees implemented as a unique data-structure with a single parallelized operation, allowing for computational efficiency without requiring any storage redundancy. We refer to the shared code for further details. As with our hyperbolic PPO extensions, we also optimize the final layer's hyperbolic weights with Riemannian Adam, keeping the same parameters as for the Adam optimizer used in the other Euclidean layers.

Table 6: Rainbow DQN hyper-parameters changes for the Atari 100K benchmark

| Rainbow DQN Atari 100K training hyper-parameters | |
| --- | --- |
| Stacked input frames | 4 |
| Batch size | 32 |
| Minimum data before training | 1600 steps |
| Network updates per step | 2 |
| Update target network every | 1 env. steps |
| $\epsilon$-greedy exploration schedule | 1→0.01 in 20K steps |

The characteristics of the Atari 100K benchmark are severely different from Procgen, given by the lack of parallelized environments and the $250\times$ reduction in total training data. Hence, we make a minimal set of changes to the training loop hyper-parameters of our Rainbow DQN implementation to ensure effective learning, as detailed in Table 6. These are based on standard practices employed by off-policy algorithm evaluating on the Atari 100K benchmark (Van Hasselt et al., 2019; Laskin et al., 2020a; Yarats et al., 2021a) and were not tuned for our specific implementation.

### C.4  EXPERIMENTAL RECORDINGS

By default, we repeat each experiment with five random seeds. To visualize the evolution of relevant values using performance curves, we collect measurements from each algorithm in five evaluation rollouts every $64 \times 1024$ frames (64 is the number of parallel environments used in Procgen). Hence, the reported values correspond to the mean results over all the collected seeds and runs, while the shaded regions correspond to the standard deviation between the mean results from different random seeds. For the tabular data illustrating final performance, we perform ten additional evaluation rollouts at the end of the specified training allowance. The reported uncertainty again represents the standard deviation between the mean results from different seeds.

### C.5  CURRENT LIMITATIONS

We identify three main current limitations of the proposed implementation. First, stabilizing hyperbolic representations with S-RYM inherently constrains the expressivity of the Euclidean subnetwork encoder model ($f_E$) to be 1-Lipschitz. This loss of expressivity might hinder the network's ability to learn complex representations, preventing our hyperbolic framework to achieve its full potential. Second, the training and evaluation time of our hyperbolic agents are consistently higher than their Euclidean counterparts. Our hyperbolic PPO implementation takes on average 4.27 seconds to collect rollouts and train for three epochs, and takes 0.961 seconds to collect a full episode of experience at the end of training. In contrast, our Euclidean baseline takes 3.69 seconds for training (16% speedup) and 0.854 seconds for evaluation (13% speedup). We find this slowdown is mainly due to the power iteration procedure performed to apply spectral normalization in S-RYM and, to a lesser extent, an overhead when computing and backpropagating through hyperbolic operations. Third, our algorithm utilizes a model of hyperbolic space with fixed negative curvature to build representations of the whole state space. However, as different RL problems might have considerably different structures, we believe that any fixed curvature might not always yield the most appropriate inductive bias. To this end, recent work showed potential benefits in using mixed curvature latent spaces and even learning the curvature parameter for unsupervised tasks (Skopek et al., 2019). We hope these limitations will be addressed in future work, further studying how differential geometry can be used to empower RL.

## D  EXTENDED RESULTS AND COMPARISONS

In this Section, we provide detailed per-environment Rainbow DQN Procgen results that were omitted from the main text due to space constraints. For both Rainbow DQN and PPO. We also compare the performance improvements from the integration of our deep hyperbolic representations with the reported improvements from recent state-of-the-art (SotA) algorithms, employing one or several orthogonal domain-specific practices. In Appendix E.3, we provide examples empirically validating

Table 7: Detailed performance comparison for the Rainbow DQN algorithm on the full Procgen benchmark. We *train* for a total of 25M steps on 200 training levels and *test* on the full distribution of levels. We report the mean returns, the standard deviation, and relative improvements from the original Rainbow DQN baseline over 5 random seeds.

| Task\Algorithm | Rainbow DQN | | Rainbow DQN + data aug. | | Rainbow DQN + S-RYM | | Rainbow DQN + S-RYM, 32 dim. | |
|---|---|---|---|---|---|---|---|---|
| **Levels distribution** | *train/test* | | *train/test* | | *train/test* | | *train/test* | |
| *bigfish* | 23.17±4 | 15.47±2 | 19.61±4 (-15%) | 17.39±4 (+12%) | 27.61±0 (+19%) | **23.03±2 (+49%)** | 30.85±2 (+33%) | 22.37±2 (+45%) |
| *bossfight* | 7.17±1 | 7.29±1 | 6.22±1 (-13%) | 6.97±1 (-4%) | 9.41±1 (+31%) | 7.75±1 (+6%) | 8.21±1 (+15%) | **8.71±1 (+20%)** |
| *caveflyer* | 7.00±1 | 3.92±1 | 7.59±0 (+8%) | 5.36±1 (+37%) | 6.39±1 (-9%) | 3.11±1 (-21%) | 6.45±1 (-8%) | **5.46±1 (+39%)** |
| *chaser* | 3.09±1 | 2.31±0 | 2.89±0 (-6%) | 2.61±1 (+13%) | 4.03±1 (+30%) | **3.65±1 (+58%)** | 3.78±0 (+23%) | 3.29±0 (+43%) |
| *climber* | 3.68±1 | 1.73±1 | 2.57±1 (-30%) | 2.36±1 (+36%) | 3.91±0 (+6%) | 2.39±0 (+38%) | 4.80±2 (+31%) | **3.00±0 (+73%)** |
| *coinrun* | 5.56±1 | 4.33±1 | 3.22±1 (-42%) | 3.00±1 (-31%) | 5.20±0 (-6%) | 5.07±1 (+17%) | 6.00±1 (+8%) | **6.33±1 (+46%)** |
| *dodgeball* | 7.42±1 | 4.67±1 | 8.91±1 (+20%) | **6.96±1 (+49%)** | 6.07±1 (-18%) | 3.60±1 (-23%) | 6.89±1 (-7%) | 5.31±1 (+14%) |
| *fruitbot* | 21.51±3 | 16.94±2 | 22.29±2 (+4%) | 20.53±3 (+21%) | 20.31±1 (-6%) | 20.30±1 (+20%) | 22.81±1 (+6%) | **21.87±2 (+29%)** |
| *heist* | 0.67±0 | 0.11±0 | 1.67±0 (+150%) | **0.67±0 (+500%)** | 1.27±0 (+90%) | 0.40±0 (+260%) | 0.93±1 (+40%) | 0.47±0 (+320%) |
| *jumper* | 5.33±1 | 3.11±0 | 4.22±0 (-21%) | 2.78±1 (-11%) | 4.78±1 (-10%) | 2.44±1 (-21%) | 5.53±1 (+4%) | **3.47±1 (+11%)** |
| *leaper* | 1.78±1 | 2.56±1 | 6.11±1 (+244%) | **5.11±1 (+100%)** | 2.00±1 (+13%) | 1.00±0 (-61%) | 0.80±0 (-55%) | 0.53±0 (-79%) |
| *miner* | 2.22±1 | **2.33±0** | 1.89±0 (-15%) | 1.33±0 (-43%) | 2.40±0 (+8%) | 1.40±0 (-40%) | 2.73±1 (+23%) | 2.00±0 (-14%) |
| *maze* | 2.01±0 | 0.67±0 | 2.07±0 (+3%) | **1.58±1 (+137%)** | 1.91±0 (-5%) | 0.93±0 (+40%) | 1.97±0 (-2%) | 0.92±0 (+38%) |
| *ninja* | 3.33±0 | 2.33±1 | 3.44±1 (+3%) | 2.56±1 (+10%) | 3.33±1 (+0%) | 2.11±0 (-10%) | 3.73±1 (+12%) | **3.33±1 (+43%)** |
| *plunder* | 8.69±0 | **6.28±1** | 6.06±1 (-30%) | 5.30±1 (-16%) | 7.33±1 (-16%) | 5.93±1 (-5%) | 7.11±1 (-18%) | 5.71±1 (-9%) |
| *starpilot* | 47.83±6 | 42.42±1 | 51.79±3 (+8%) | 46.23±5 (+9%) | 57.64±2 (+21%) | **55.86±3 (+32%)** | 59.94±1 (+25%) | 54.77±3 (+29%) |
| Average norm. score | 0.2679 | 0.1605 | 0.2698 (+1%) | 0.2106 (+31%) | 0.2774 (+4%) | 0.1959 (+22%) | **0.3097 (+16%)** | **0.2432 (+51%)** |
| Median norm. score | 0.1856 | 0.0328 | 0.1830 (-1%) | 0.1010 (+208%) | 0.2171 (+17%) | 0.0250 (-24%) | **0.2634 (+42%)** | **0.1559 (+376%)** |
| # Env. improvements | 0/16 | 0/16 | 8/16 | 11/16 | 8/16 | 9/16 | **11/16** | **13/16** |

that hyperbolic representations provide mostly complementary benefits and are compatible with different domain-specific practices, potentially yielding even further performance gains. Finally, we provide additional qualitative 2-dimensional visualizations of learned trajectory representation and $\delta$-hyperbolicity recordings also for our regularized hyperbolic PPO algorithm.

## D.1 RAINBOW DQN PROCGEN RESULTS

As shown in Table 7, our hyperbolic Rainbow DQN with S-RYM appears to yield conspicuous performance gains on the majority of the environments. Once again, we find that reducing the dimensionality of the representations to 32 provides even further benefits, outperforming the Euclidean baseline in 13 out of 16 environments. This result not only highlights the efficiency of hyperbolic geometry to encode hierarchical features, but also appears to validate our intuition about the usefulness of regularizing the encoding of non-hierarchical and potentially spurious information. While still inferior to our best hyperbolic implementation, data augmentations seem to have a greater overall beneficial effect when applied to Rainbow DQN rather than PPO. We believe this result is linked with recent literature (Cetin et al., 2022) showing that data-augmentation also provides off-policy RL with an auxiliary regularization effect that stabilizes temporal-difference learning.

## D.2 SOTA COMPARISON ON PROCGEN

In Table 8 we compare our best hyperbolic PPO agent with the reported results for the current SotA Procgen algorithms from Raileanu & Fergus (2021). All these works propose domain-specific practices on top of PPO (Schulman et al., 2017), designed and tuned for the Procgen benchmark: Mixture Regularization (MixReg) (Wang et al., 2020), Prioritized Level Replay (PLR) (Jiang et al., 2021), Data-regularized Actor-Critic (DrAC) (Raileanu et al., 2020), Phasic Policy Gradient (PPG) (Cobbe et al., 2021), and Invariant Decoupled Advantage Actor Critic (Raileanu & Fergus, 2021). Validating our implementation, we see that our Euclidean PPO results closely match the previously reported ones, lagging severely behind all other methods. In contrast, we see that introducing our deep hyperbolic representations framework makes PPO outperform all considered baselines but IDAAC, attaining overall similar scores to this algorithm employing several domain-specific practices. In particular, IDAAC not only makes use of a very specialized architecture, but also introduces an auxiliary objective to minimize the correlation between the policy representations and the number of steps until task-completion. Raileanu & Fergus (2021) found this measure to be an effective heuristic correlating with the occurrence of overfitting in many Procgen environments. Moreover, we see that our hyperbolic PPO attains the best performance on 7 different environments, more than any

Table 8: Performance comparison on the test distribution of levels for our Euclidean and Hyperbolic PPO agents with the reported results of recent RL algorithms designed specifically for the Procgen benchmark.

| Task\Algorithm | PPO (Reported) | Mixreg | PLR | UCB-DrAC | PPG | IDAAC | PPO (Ours) | Hyperbolic PPO + S-RYM (Ours) |
|---|---|---|---|---|---|---|---|---|
| *bigfish* | 3.7 | 7.1 | 10.9 | 9.2 | 11.2 | 18.5 | 1.46±1 | **16.57±2 (+1037%)** |
| *bossfight* | 7.4 | 8.2 | 8.9 | 7.8 | **10.3** | 9.8 | 7.04±2 | 9.02±1 (+28%) |
| *caveflyer* | 5.1 | 6.1 | **6.3** | 5 | 7 | 5 | 5.86±1 | 5.20±1 (-11%) |
| *chaser* | 3.5 | 5.8 | 6.9 | 6.3 | **9.8** | 6.8 | 5.89±1 | 7.32±1 (+24%) |
| *climber* | 5.6 | 6.9 | 6.3 | 6.3 | 2.8 | **8.3** | 5.11±1 | 7.28±1 (+43%) |
| *coinrun* | 8.6 | 8.6 | 8.8 | 8.9 | 8.9 | **9.4** | 8.25±0 | 9.20±0 (+12%) |
| *dodgeball* | 1.6 | 1.7 | 1.8 | 4.2 | 2.3 | 3.2 | 1.87±1 | **7.14±1 (+281%)** |
| *fruitbot* | 26.2 | 27.3 | 28 | 27.6 | 27.8 | 27.9 | 26.33±2 | **29.51±1 (+12%)** |
| *heist* | 2.5 | 2.6 | 2.9 | 3.5 | 2.8 | 3.5 | 2.92±1 | **3.60±1 (+23%)** |
| *jumper* | 5.9 | 6 | 5.8 | 6.2 | 5.9 | **6.3** | 6.14±1 | 6.10±1 (-1%) |
| *leaper* | 4.9 | 5.3 | 6.8 | 4.8 | **8.5** | 7.7 | 4.36±2 | 7.00±1 (+61%) |
| *maze* | 5.5 | 5.2 | 5.5 | 6.3 | 5.1 | 5.6 | 6.50±0 | **7.10±1 (+9%)** |
| *miner* | 8.4 | 9.4 | 9.6 | 9.2 | 7.4 | 9.5 | 9.28±1 | **9.86±1 (+6%)** |
| *ninja* | 5.9 | 6.8 | **7.2** | 6.6 | 6.6 | 6.8 | 6.50±1 | 5.60±1 (-14%) |
| *plunder* | 5.2 | 5.9 | 8.7 | 8.3 | 14.3 | **23.3** | 6.06±3 | 6.68±0 (+10%) |
| *starpilot* | 24.9 | 32.4 | 27.9 | 30 | **47.2** | 37 | 26.57±5 | 38.27±5 (+44%) |
| **Average norm. score** | 0.3078 | 0.3712 | 0.4139 | 0.3931 | 0.4488 | **0.5048** | 0.3476 | 0.4730 (+36%) |
| **Median norm. score** | 0.3055 | 0.4263 | 0.4093 | 0.4264 | 0.4456 | **0.5343** | 0.3457 | 0.4705 (+36%) |

Table 9: Performance comparison for our Euclidean and Hyperbolic Rainbow DQN agents with the reported results of recent RL algorithms designed specifically for the Atari 100K benchmark.

| Task\Algorithm | Random | Human | DER | OTRainbow | CURL | DrQ | SPR | Rainbow DQN (Ours) | Rainbow DQN + S-RYM (Ours) |
|---|---|---|---|---|---|---|---|---|---|
| *Alien* | 227.80 | 7127.70 | 739.9 | **824.7** | 558.2 | 771.2 | 801.5 | 548.33 | 679.20 (+41%) |
| *Amidar* | 5.80 | 1719.50 | **188.6** | 82.8 | 142.1 | 102.8 | 176.3 | 132.55 | 118.62 (-11%) |
| *Assault* | 222.40 | 742.00 | 431.2 | 351.9 | 600.6 | 452.4 | 571 | 539.87 | **706.26 (+52%)** |
| *Asterix* | 210.00 | 8503.30 | 470.8 | 628.5 | 734.5 | 603.5 | **977.8** | 448.33 | 535.00 (+36%) |
| *Bank Heist* | 14.20 | 753.10 | 51 | 182.1 | 131.6 | 168.9 | **380.9** | 187.5 | 255.00 (+39%) |
| *Battle Zone* | 2360.00 | 37187.50 | 10124.6 | 4060.6 | 14870 | 12954 | 16651 | 12466.7 | **25800.00 (+132%)** |
| *Boxing* | 0.10 | 12.10 | 0.2 | 2.5 | 1.2 | 6 | **35.8** | 2.92 | 9.28 (+226%) |
| *Breakout* | 1.70 | 30.50 | 1.9 | 9.8 | 4.9 | 16.1 | 17.1 | 13.72 | **58.18 (+370%)** |
| *Chopper Command* | 811.00 | 7387.80 | 861.8 | 1033.3 | **1058.5** | 780.3 | 974.8 | 791.67 | 888.00 (+498%) |
| *Crazy Climber* | 10780.50 | 35829.40 | 16185.3 | 21327.8 | 12146.5 | 20516.5 | **42923.6** | 20496.7 | 22226.00 (+18%) |
| *Demon Attack* | 152.10 | 1971.00 | 508 | 711.8 | 817.6 | 1113.4 | 545.2 | 1204.75 | **4031.60 (+269%)** |
| *Freeway* | 0.00 | 29.60 | 27.9 | 25 | 26.7 | 9.8 | 24.4 | **30.5** | 29.50 (-3%) |
| *Frostbite* | 65.20 | 4334.70 | 866.8 | 231.6 | 1181.3 | 331.1 | **1821.5** | 318.17 | 1112.20 (+314%) |
| *Gopher* | 257.60 | 2412.50 | 349.5 | 778 | 669.3 | 636.3 | 715.2 | 343.67 | **1132.80 (+917%)** |
| *Hero* | 1027.00 | 30826.40 | 6857 | 6458.8 | 6279.3 | 3736.3 | 7019.2 | **9453.25** | 7654.40 (-21%) |
| *Jamesbond* | 29.00 | 302.80 | 301.6 | 112.3 | 471 | 236 | 365.4 | 190.83 | **380.00 (+117%)** |
| *Kangaroo* | 52.00 | 3035.00 | 779.3 | 605.4 | 872.5 | 940.6 | **3276.4** | 1200 | 1020.00 (-16%) |
| *Krull* | 1598.00 | 2665.50 | 2851.5 | 3277.9 | **4229.6** | 4018.1 | 3688.9 | 3445.02 | 3885.02 (+24%) |
| *Kung Fu Master* | 258.50 | 22736.30 | **14346.1** | 5722.2 | 14307.8 | 9111 | 13192.7 | 7145 | 10604.00 (+50%) |
| *Ms Pacman* | 307.30 | 6951.60 | 1204.1 | 941.9 | **1465.5** | 960.5 | 1313.2 | 1044.17 | 1135.60 (+12%) |
| *Pong* | -20.70 | 14.60 | -19.3 | 1.3 | -16.5 | -8.5 | -5.9 | 3.85 | **11.98 (+33%)** |
| *Private Eye* | 24.90 | 69571.30 | 97.8 | 100 | **218.4** | -13.6 | 124 | 72.28 | 106.06 (+71%) |
| *Qbert* | 163.90 | 13455.00 | 1152.9 | 509.3 | 1042.4 | 854.4 | 669.1 | 860.83 | **2702.00 (+264%)** |
| *Road Runner* | 11.50 | 7845.00 | 9600 | 2696.7 | 5661 | 8895.1 | 14220.5 | 6090 | **22256.00 (+266%)** |
| *Seaquest* | 68.40 | 42054.70 | 354.1 | 286.9 | 384.5 | 301.2 | **583.1** | 259.33 | 476.80 (+114%) |
| *Up N Down* | 533.40 | 11693.20 | 2877.4 | 2847.6 | 2955.2 | 3180.8 | **28138.5** | 2935.67 | 3255.00 (+13%) |
| **Human Norm. Mean** | 0.000 | 1.000 | 0.285 | 0.264 | 0.381 | 0.357 | **0.704** | 0.353 | 0.686 (+94%) |
| **Human Norm. Median** | 0.000 | 1.000 | 0.161 | 0.204 | 0.175 | 0.268 | **0.415** | 0.259 | 0.366 (+41%) |
| **# Super** | N/A | N/A | 2 | 1 | 2 | 2 | **7** | 2 | 5 |

other method. Furthermore, in these environment the other Euclidean algorithms specifically struggle, again indicating the orthogonal effects of our approach as compared to traditional RL advances.

### D.3 SOTA COMPARISON ON ATARI 100K

In Table 9 we provide detailed raw results for our hyperbolic Rainbow DQN agent, comparing with the results for recent off-policy algorithms for the Atari 100K benchmark, as reported by Schwarzer et al. (2020). All the considered algorithms build on top of the original Rainbow algorithm (Hessel et al., 2018). We consider Data Efficient Rainbow (DER) (Van Hasselt et al., 2019) and Overtrained Rainbow (OTRainbow) (Kielak, 2019) which simply improve the model architectures and other training-loop hyper-parameters, for instance, increasing the number of update steps per collected environment step. We also compare with other more recent baselines that incorpo-

rate several additional auxiliary practices and data-augmentation such as Data-regularized Q (DrQ) (Yarats et al., 2021a), Contrastive Unsupervised Representations (CURL) (Laskin et al., 2020a), and Self-Predictive Representations (SPR) (Schwarzer et al., 2020). While our Euclidean Rainbow implementation attains only mediocre scores, once again we see that introducing our deep hyperbolic representations makes our approach competitive with the state-of-the-art and highly-tuned SPR algorithm. In particular, SPR makes use of several architectural advancements, data-augmentation strategies from prior work, and a model-based contrastive auxiliary learning phase. Also on this benchmark, our hyperbolic agent attains the best performance on 8 different environments, more than any other considered algorithm.

### D.4 2-DIMENSIONAL REPRESENTATIONS PERFORMANCE AND INTERPRETATION

To visualize and allow us to interpret the structure of the learned representations, we analyze our hyperbolic PPO agents using only *two dimensions* to model the final latent representations. As mentioned in Section 4 and shown in Table 10, we find even this extreme implementation to provide performance benefits on the test levels over Euclidean PPO. Furthermore, the generalization gap with the training performance is almost null in three out of the four considered environments. As the 2-dimensional representation size greatly con-

Table 10: Performance of 2-dimensional hyperbolic PPO as compared to the original PPO algorithm.

| Task\Algorithm | PPO + S-RYM, 2 dim. | |
|---|---|---|
| **Levels distribution** | *train/test* | |
| *bigfish* | 5.65±4 (+52%) | 2.34±3 (+60%) |
| *dodgeball* | 2.62±0 (-48%) | 2.36±1 (+26%) |
| *fruitbot* | 27.18±4 (-10%) | 25.75±1 (-2%) |
| *starpilot* | 30.27±3 (-1%) | 29.72±6 (+12%) |

strains the amount of encoded information, these results provide further validation for the affinity of hyperbolic geometry to effectively prioritize features useful for RL. We then observe how these 2-dimensional hyperbolic latent representations evolve within trajectories, mapping them on the Poincaré disk and visualizing the corresponding input states. As summarized in Section 4, we observe a recurring *cyclical* behavior, where the magnitude of the representations monotonically increases within subsets of the trajectory as more obstacles and/or enemies appear. Together with Figure 10 (on the *bigfish* environment), we provide additional qualitative visualizations of this phenomenon in Figure 12 (on the *starpilot* (A), *dodgeball* (B), and *fruitbot* (C) environments). These plots compare the representations of on-policy states sampled at constant intervals within a trajectory, every 15 timesteps, and deviations from executing 15 timesteps of random behavior after resetting the environment to the previous on-policy state. We observe the state representations form tree-like branching structures, somewhat reflecting the tree-like nature of MDPs. Within the subtrajectories in *starpilot* and *fruitbot*, we find that the magnitudes in the on-policy trajectory tend to grow in the direction of the Value function's *gyroplane*'s normal. Intuitively, this indicates that as new elements appear (e.g., new enemies in *starpilot*), the agent recognizes a larger opportunity for rewards (e.g., from defeating them) and also that it now requires a much finer level of control for optimality. This is because as magnitudes increase, the signed distances with the policy gyroplanes will also grow exponentially, and so will the value of the different action logits, decreasing the policy's entropy. In contrast, the magnitudes of the state representations following the random deviations grow in directions with considerably larger orthogonal components to the Value gyroplane's normal. This still reflects the higher precision required for optimal decision-making, as magnitudes still increase, but also the higher uncertainty to obtain future rewards from these less optimal states. As opposed to the rest of the environments, in *dodgeball* all enemies and other elements are already present from the first time-step of a trajectory. Furthermore, our 2-dimensional hyperbolic agent appears to lack the representation power to recover good performance. These two properties make the observed branching phenomenon less accentuated, with the magnitude of both random and on-policy transitions changing in mostly orthogonal directions to the gyroplane's normal.

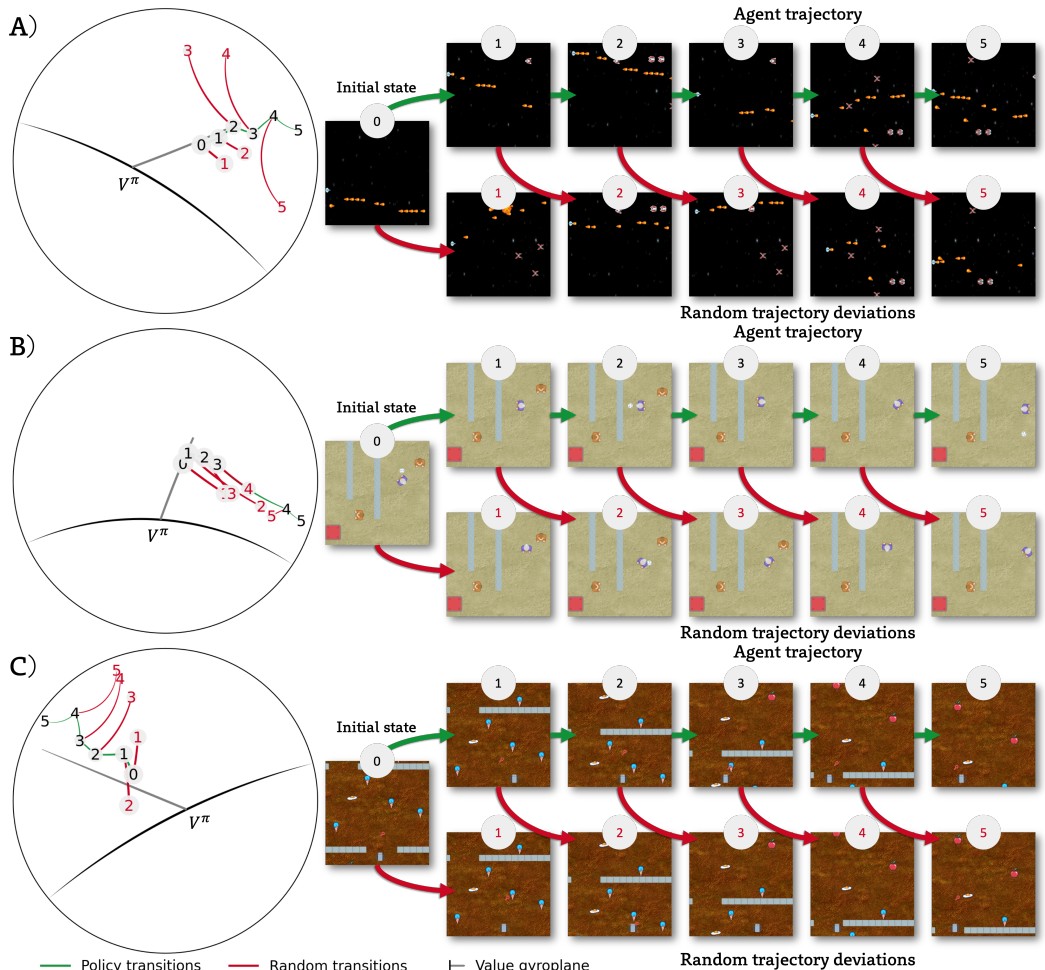

Figure 12: Visualization of 2-dimensional hyperbolic embeddings in the *starpilot* (**A**), *dodgeball* (**B**), and *fruitbot* (**C**) Procgen environment. We sub-sample states from recorded agent trajectories every 15 timesteps. We show the evolution of the hyperbolic latent representations following the recorded policy transitions as compared to random transitions collected by resetting the environments from each state and executing a random policy for the same 15 timesteps.

## D.5 $\delta$ HYPERBOLICITY

We repeat the experiment in Figure 4, collecting the relative $\delta$-hyperbolicity of the latent representations space produced by our regularized Hyperbolic PPO agent throughout training. Our analysis assumes that as different RL tasks likely require encoding different amounts of hierarchically-structured information, relative changes in $\delta$-hyperbolicity should be more informative than its overall scale. Furthermore, we would like to emphasize that we tractably estimate $\delta_{rel}$ using the efficient but approximate algorithm from Fournier et al. (2015) and that, in practice, there are many sources of noise that affect RL optimization. Both these factors inevitably add uninformative noise to the latent representations and our hyperbolicity recordings, which likely affects many of the small local observed changes in $\delta_{rel}$.

We visualize the evolution of the $\delta$-hyperbolicity of our hyperbolic PPO agent regularized with S-RYM in Figure 13. As we would expect, using a hyperbolic latent space yields latent representations with significantly lower values of $\delta_{rel}$, as compared to standard PPO, implying they possess an increased hierarchical tree-like structure. We observe this consistently for all considered tasks and during all stages of the RL training process. This difference is particularly evident at initialization where $\delta_{rel} \approx 0.2$ for our hyperbolic PPO while $\delta_{rel} \approx 0.4$ for a standard PPO agent, reflecting

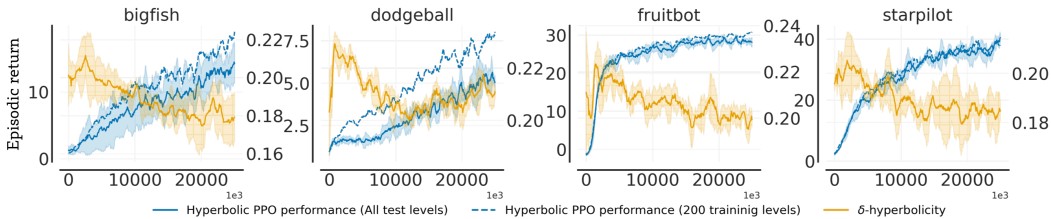

Figure 13: Performance and relative $\delta$-hyperbolicity of the final latent representations of our hyperbolic PPO agent regularized with S-RYM.

how the properties of hyperbolic space facilitate capturing hierarchical relationships even with no training. Interestingly, in the *dodgeball* environment, the recorded value of $\delta_{rel}$ for our Hyperbolic PPO experiences some considerable fluctuations in some of the very initial and later training stages. Analogously to our results in Figure 4, it appears that during the iterations where $\delta_{rel}$ increases or attains its higher values, test performance grows significantly slower. Overall, *dodgeball* is also the environment with the largest generalization gap relative to the hyperbolic agent's training performance.

## E  FURTHER EXPERIMENTS AND ABLATION STUDIES

In this section, we further analyze the properties of our hyperbolic RL framework and its implementation, through additional experiments and ablations. We focus on our hyperbolic PPO algorithm and four representative tasks from the Procgen benchmark.

### E.1  S-RYM'S COMPONENTS CONTRIBUTION

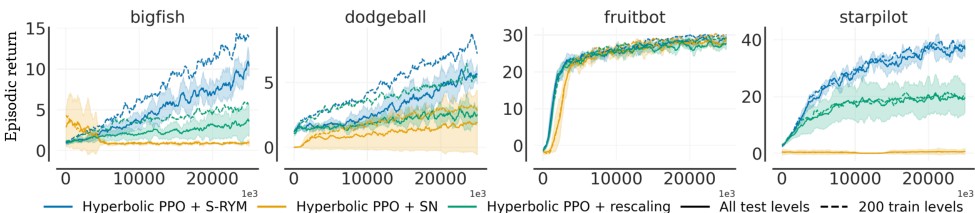

Figure 14: Performance ablating either spectral normalization or rescaling from our Hyperbolic PPO agent stabilized with S-RYM.

Our proposed spectrally-regularized hyperbolic mappings (S-RYM) relies on two main distinct components: spectral normalization and rescaling. As described in Section 3, we design our deep RL models to produce a representation by applying traditional neural network layers in Euclidean space $x_E = f_E(s)$. Before the final linear layer $f_H$, we then use an exponential map from the origin of the Poincaré to yield a final representation in hyperbolic space $x_H = \exp_0^1(x_E)$. As shown by Lin et al. (2021), applying spectral normalization to the layers of $f_E$ regulates both the values and gradients similarly to LeCun initialization (LeCun et al., 2012). Hence, we make the regularization approximately dimensionality-invariant by rescaling $x_E \in \mathbb{R}^n$, simply dividing its value by $\sqrt{n}$. In Figure 14, we show the results from ablating either component from S-RYM. From our results, both components seem crucial for performance. As removing spectral normalization simply recovers the unregularized hyperbolic PPO implementation with some extra rescaling in the activations, its performance is expectedly close to the underwhelming performance of our naive implementations in Figure 5. Removing our dimensionality-based rescaling appears to have an even larger effect, with almost no agent improvements in 3 out of 4 environments. The necessity of appropriate scaling comes from the influence the representations magnitudes have on optimization. When applying spectral normalization, the dimensionality of the representations directly affects its expected magnitude. Thus, high-dimensional latents will result in high-magnitude representations, making it challenging to optimize for appropriate angular layouts in hyperbolic space (Nickel & Kiela, 2017;

Ganea et al., 2018) and making the gradients of the Euclidean network parameters stagnate (Guo et al., 2022). These issues cannot even be alleviate with appropriate network initialization, since the magnitudes of all weights will be rescaled by the intruduced spectral normalization.

## E.2 REPRESENTATION SIZE

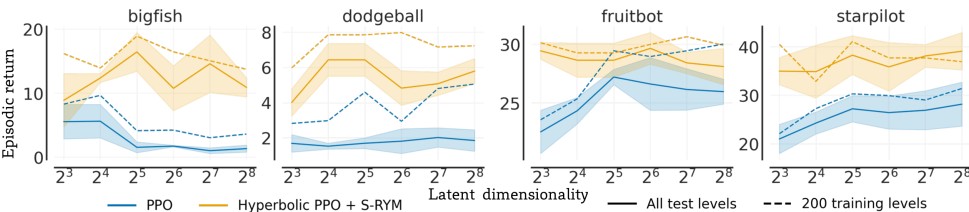

Figure 15: Final performance comparison between PPO agents with Euclidean and hyperbolic representations with different dimensionalities.

In Figure 15, we show the final train and test performance attained by our Euclidean and hyperbolic PPO agents with different dimensionalities for their final latent representations. We collect results on a log scale $2^n$ with $n \in \{3, 4, 5, 6, 7, 8\}$, i.e., ranging from $2^3 = 8$ to $2^8 = 256$ latent dimensions. Integrating our hyperbolic representations framework with PPO boosts performance across all dimensionalities. Moreover, in 3/4 environments we see both train and test performance of the Euclidean PPO agent considerably dropping as we decrease the latent dimensions. In contrast, the performance of hyperbolic PPO is much more robust, even attaining some test performance gains from more compact representations. As described in Section 2, Euclidean representations require high dimensionalities to encode hierarchical features with low distortion (Matoušek, 1990; Gupta, 1999), which might explain their diminishing performance. Instead, as hyperbolic representations do not have such limitation, lowering the dimensionality should mostly affect their ability of encoding non-hierarchical information, which we believe to counteract the agent's tendency of overfitting to the limited distribution of training levels and observed states.

## E.3 COMPATIBILITY WITH ORTHOGONAL PRACTICES

Introducing hyperbolic geometry to model the representations of RL agents is fundamentally orthogonal to most recent prior advances. Thus, we validate the compatibility of our approach with different methods also aimed at improving the performance and generalization of PPO.

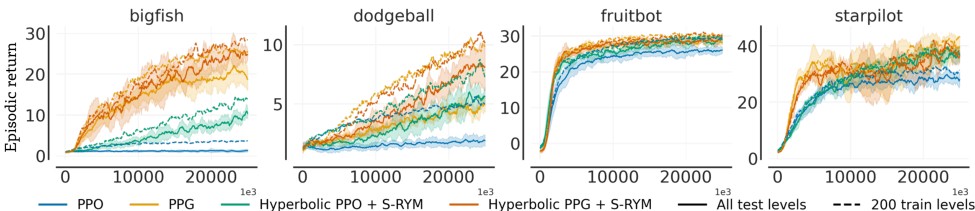

Figure 16: Performance comparison from integrating the advances from the PPG algorithm our hyperbolic reinforcement learning framework.

**Phasic Policy Gradient (PPG).** We re-implement this recent PPO extension designed by Cobbe et al. (2021) specifically for the Procgen benchmark. PPG adds non-trivial algorithmic and computational complexity, by performing two separate optimization phases. In the first phase, it optimizes the same policy and value optimization objective as in PPO, utilizing the latest on-policy data. In the second phase, it utilizes a much larger buffer of past experience to learn better representations in its policy model via an auxiliary objective, while avoiding forgetting with an additional behavior cloning weighted term. The two phases are alternated infrequently after several training epochs. Once again, we incorporate our hyperbolic representation framework on top of PPG without any additional tuning. In Figure 16, we show the results from adding our deep hyperbolic representation

framework to PPG. Even though PPG's performance already far exceeds PPO, hyperbolic representations appear to have similar effects on the two algorithms, with performance on the 200 training levels largely invaried, and especially notable test performance gains on the bigfish and dodgeball environments. Hence, in both PPO and PPG, the new prior induced by the hyperbolic representations appears to largely reduce overfitting to the observed data and achieve better generalization to unseen conditions. Our approach affects RL in an orthogonal direction to most other algorithmic advances, and our results appear to confirm the general compatibility of its benefits.

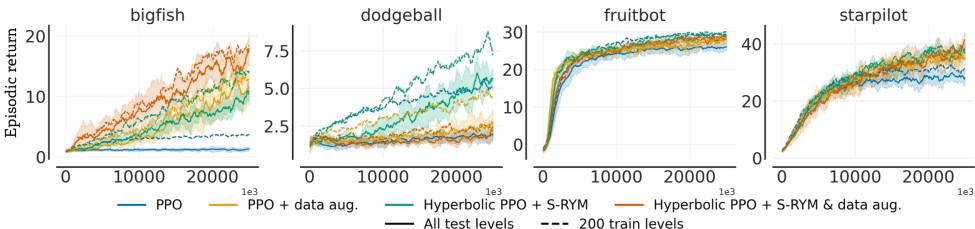

Figure 17: Performance comparison from integrating data augmentation with the Euclidean and hyperbolic PPO agents.

**Data augmentation.** Finally, we also test introducing data augmentation to our Hyperbolic PPO implementation. We consider the same popular random shifts from Yarats et al. (2021a), evaluated in Section 4. We note that the problem diversity characterizing procgen makes it challenging for individual hand-designed augmentations to have a generally beneficial effect, with different strategies working best in different environments (Raileanu et al., 2020). In fact, applying random shifts to PPO appears to even hurt performance on a considerable subset of environments (see Table 1), likely due to the agents losing information about the exact position and presence of key objects at the borders of the environment scene. This inconsistency is reflected onto the hyperbolic PPO agent. In particular, while the addition of random shifts further provides benefits on the bigfish environment, it appears to hurt performance on dodgeball. Overall, integrating our hyperbolic framework still appears considerably beneficial even for the test performance of the data-augmented agent, further showing the generality of our method.

### E.4 ENFORCING LIPSCHITZ CONTINUITY

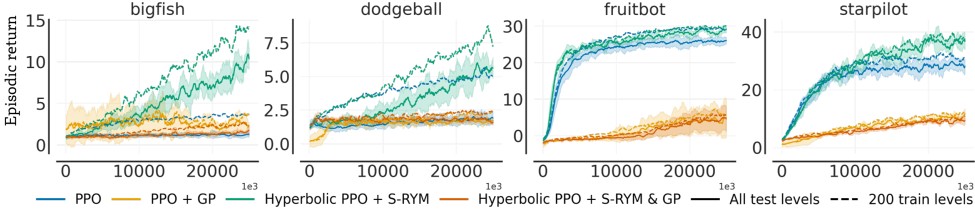

Figure 18: Performance of a standard PPO agent and the hyperbolic PPO agent stabilized with S-RYM after integrating gradient penalties (GP) (Gulrajani et al., 2017a).

S-RYM leaves the final layer of the network unregularized since there is no direct way of performing power iteration with its parameterization and we also do not want to constrain our models of the value and policy to be 1-Lipschitz. We validate that this property is not reflective of the true optimal policy and value functions, by enforcing Lipschitz continuity with gradient penalties (GP) (Gulrajani et al., 2017a). We apply GP on top of both our Hyperbolic PPO with S-RYM and standard PPO. Our results in Figure E.4 appear to validate our hypothesis by showing that enforcing either the hyperbolic or Euclidean PPO models to be 1-Lipschitz makes performance collapse across all environments.

