# OpenReview forum: "Hyperbolic Deep Reinforcement Learning"
_ICLR.cc/2023/Conference — ICLR 2023 notable top 25%_

### Official Review · Reviewer_56hh · 2022-10-22

**Confidence:** 2
**Correctness:** 4
**Technical Novelty And Significance:** 3
**Empirical Novelty And Significance:** 3
**Recommendation:** 6

**Clarity, Quality, Novelty And Reproducibility:**

* Clarity: Good.
* Quality and Novelty: The quality is pretty good and the proposed method is novel.
* Reproducibility: The source code is provided, while I have not checked it.

**Strength And Weaknesses:**

Strength:
* The paper is clearly written and easy to follow.
* The idea of learning latent representations using hyperbolic space for RL is novel, and successfully adapting hyperbolic deep learning methods to RL is non-trivial.
* The empirical performance improvements on Procgen and Atari are significant.
* Visualizations of 2-dimensional hyperbolic embeddings (Figure 10 and 12) are very intriguing.

Questions:
* In general, the state relationship is more like a directed graph than a tree. I am curious how this would fit into the hyperbolic latent learning framework in this paper?
* Can the authors provide more details on how "Euclidean PPO + S-RYM" in Figure 7 is implemented?
* It would be better if the authors can make more visualization (like Figure 10) for other games in Procgen.

Minor issues:
* Typo: below Table 4, "common practice win on-policy methods", win -> in?

**Summary Of The Paper:**

This paper proposes to model the latent state representation in RL with hyperbolic space. They find that naively applying existing hyperbolic deep learning methods is not helpful, and introduce two techniques (spectral normalization and feature rescaling) to address the issues. The integrated method (S-RYM) is evaluated on Procgen and Atari-100K benchmarks, effectively boosting the performance compared to baselines.

**Summary Of The Review:**

In summary, this paper identifies and resolves the issues when applying hyperbolic deep learning methods to RL. Modelling latent representations with hyperbolic space naturally fits the hierarchical relationship between states in RL. Extensive experiments on Procgen and Atari-100K demonstrate the effectiveness of hyperbolic RL.

---

> ### Author Response · Authors · 2022-11-13
> **Responses to Reviewer 56hh**
>
> **Questions:**
>
>
> > 1. In general, the state relationship is more like a directed graph than a tree. I am curious how this would fit into the hyperbolic latent learning framework in this paper?
>
> One of the defining characteristics of hyperbolic geometry is exponential volume growth (as noted in Section 1, bottom of first page). While trees are a particular example of graphs reflecting such characteristics, there are also many other examples (e.g., scale-free graphs). For this reason, hyperbolic space is a popular choice to embed large directed graphs arising in different settings as these structures often have exponential volume growth (see e.g., [1], [2]). While it is possible that such structures do not include tree-like branching, this would automatically result in a very simple problem setting for RL, where the evolution of the environment is deterministic and unaffected by the agent's actions. In such conditions, either hyperbolic or Euclidean space would be trivially able to encode the state relationship with arbitrarily low distortion. However, as implied by the Reviewer's question, we do agree that different RL problems might have a considerably different hierarchical structure. Hence, using our fixed-curvature model of hyperbolic space might not always provide the most optimal inductive bias to deep RL. To address this potential limitation, future work could consider using mixed curvature latent spaces and even learning the curvature parameter, as proposed in [3]. Following the Reviewer's interest, **we added a discussion of this potential limitation to Section C.5 of the Appendix**.
>
>
> > 2. Can the authors provide more details on how "Euclidean PPO + S-RYM" in Figure 7 is implemented?
>
> The Euclidean PPO + S-RYM baseline is implemented by applying the stabilization procedure of S-RYM (i.e., spectral normalization and rescaling of the final latent representations) to a standard PPO agent entirely parameterized in Euclidean space. This baseline serves to validate that it is the inductive bias from hyperbolic space that provides the main performance benefits and that the main role of S-RYM is to counteract the observed instabilities in our integrations of hyperbolic geometry with RL. In preliminary experiments, we also tried running Euclidean PPO with only each component of S-RYM individually, and we similarly observed that neither spectral normalization nor rescaling provides significant improvements to Euclidean PPO. Following the Reviewer's question, **we tried making the implementation of this baseline clearer in the last paragraph of Section 3.3**: "To validate that the performance benefits are due to hyperbolic geometry of the latent space, we also evaluate a baseline using SN and rescaling in Euclidean space, which fails to attain consistent improvements."
>
>
> > 3. It would be better if the authors can make more visualization (like Figure 10) for other games in Procgen.
>
> Following the Reviewer's suggestion, **we expanded Figure 12 in Appendix D.4, adding additional qualitative visualizations and analysis of the learned 2-dimensional representations** also for the dodgeball and fruitbot environments. In particular, we find that in the bigfish, starpilot, and fruitbot environments, the magnitude of the representations tends to consistently grow in the described tree-like pattern with the appearance of new enemies and other elements. However, in the dodgeball environment, all enemies and other elements are already present from the first time-step of a trajectory. Furthermore, our 2-dimensional hyperbolic agent appears to lack the representation power to recover good performance for this particular task. These two properties make the observed branching phenomenon less accentuated, with the magnitude of both random and on-policy transitions changing in mostly orthogonal directions to the gyroplane's normal. We believe these observations might also be connected to the higher $\delta$-hyperbolicity recorded in this environment by both the Euclidean agent (Figure 4) and the hyperbolic agent (Figure 13).
>
>
> **Minor issues:**
>
>
> > - Typo: below Table 4, "common practice win on-policy methods", win -> in?
>
> We **fixed this typo** in our latest revision.
>
>
> **References**
>
> [1] Munzner, Tamara. "H3: Laying out large directed graphs in 3D hyperbolic space." Proceedings of VIZ'97: Visualization Conference, Information Visualization Symposium and Parallel Rendering Symposium. IEEE, 1997.
>
> [2] Suzuki, Ryota, Ryusuke Takahama, and Shun Onoda. "Hyperbolic disk embeddings for directed acyclic graphs." International Conference on Machine Learning. PMLR, 2019.
>
> [3] Skopek, Ondrej, Octavian-Eugen Ganea, and Gary Bécigneul. "Mixed-curvature Variational Autoencoders." International Conference on Learning Representations. 2019.

---

### Official Review · Reviewer_4ZkJ · 2022-10-26

**Confidence:** 4
**Correctness:** 4
**Technical Novelty And Significance:** 4
**Empirical Novelty And Significance:** 4
**Recommendation:** 10

**Clarity, Quality, Novelty And Reproducibility:**

**Clarity:**
The paper is very clear and well-written. Several helpful figures are included that make it easier to gain intuition about the concepts introduced. Thank you, authors!

**Quality:**
I have a few concerns about some of the claims made in the paper (detailed above), but overall the work seems well-done to me.

**Novelty:**
As far as I am aware, hyperbolic representations have not been used in *deep* RL before, but I am also not a specialist in this area.

**Reproducibility:**
Enough details are included in the paper and appendices to make reproducing the results possible.

**Strength And Weaknesses:**

**Strengths:**
+ Very well-written and clear; the analogy between hyperbolic spaces and tree structures was very helpful for gaining intuition, as were the examples provided throughout (Figures 2 and 3 in particular).
+ The introduction does a good job of arguing for hyperbolic representations for a variety of reasons.
+ An ablation study performed on the two components (spectral normalization of the euclidean sub-network and rescaling of its output) of the proposed method S-RYM showing they are both necessary.
+ Broad empirical demonstrations that result in fairly consistent performance improvements.

**Potential weaknesses/questions:**
1. In Figure 4, the relationship between $\delta$-hyperbolicity and the train-test gap seems very questionable. In all of the plots, the train-test gap grows while $\delta$-hyperbolicity is shrinking. Additionally, in the fruitbot plot $\delta$-hyperbolicity increases several times with no apparent change in the train-test gap. Furthermore, the scale of the $\delta$-hyperbolicity is the same between representations that learned to successfully solve the task (fruitbot and starpilot) and those that failed to generalize (bigfish and dodgeball), suggesting all of the representations are of a similar level of hyperbolicity. Doesn't that suggest that something else is the cause of the failure to generalize? It definitely does not validate the hypothesis.
1. Much of the reasoning in the first part of Section 3.3 seems very speculative. It would be good to adjust the language to better convey that this explanation is a hypothesis.
1. How do standard neural net layers yield Euclidean velocity vectors? This detail seemed to be glossed over.
1. Some of the figures and most of the plots are too small to read easily. Consider removing the depictions of the environments, and moving the legends to be horizontally laid out underneath the plots. That should allow for larger plots without too much vertical space being used.
1. The colour scheme is not accessible to people with colourblindness, and the size of the lines in the legends are too small to easily see their colour.
1. What are the shaded regions in the plots? Confidence intervals? I wasn't able to find this information.
1. "our instabilities appear to occur in the gradients from the hyperbolic representations". This was not clear to me from Figure 6. Is there some reason to not use spectral normalization on the hyperbolic representations? Did it hurt performance?
1. Do the representations learned using S-RYM actually exhibit increased $\delta$-hyperbolicity over the base algorithms? I'm very curious about this, but couldn't find this information in the appendices.
1. What are some of the shortcomings of the proposed algorithm? I could not find any discussion of limitations.

**Minor comments/questions:**
- Bottom of page 2: duplicate "upon".
- Bottom of page 3: "of the form in Equation 6" would be clearer.
- Figure 6 appears before Figure 5.
- What does the dashed line represent in Figure 5?
- Start of related work section: "objectives appears has been shown prone to overfitting"
- The legend in Figure 8 seems not quite right. The legend indicates dashed lines represent performance on the 200 training levels, but the plots involve a varying number of training levels.

**Summary Of The Paper:**

The paper introduces hyperbolic representations to deep reinforcement learning, with the intuition that hyperbolic space should be better for representing the sorts of hierarchical relationships between subsequent states found in deep RL. However, a straightforward implementation built on top of PPO does not seem to improve performance, and in fact results in worse performance on some environments. The paper hypothesizes that this is due to optimization difficulties introduced by the hyperbolic layer, and proposes using spectral normalization and rescaling of the hyperbolic layer's outputs to alleviate the optimization difficulties. The resulting module improves performance when combined with both PPO and Rainbow DQN on a broad set of benchmarks. Extensive empirical investigations suggest that hyperbolic representations could be a good basis for building future deep RL algorithms.

**Summary Of The Review:**

I recommend accepting the paper for publication; it is very clearly written, proposes a new approach, shows a general way to get it to work with existing algorithms, and demonstrates the benefits on multiple benchmarks. It would be nice if my concerns were addressed, but even without doing so the paper should probably be published anyways.

**Update:** The authors have addressed all of my concerns satisfactorily, and I can now strongly recommend accepting the paper.

---

> ### Author Response · Authors · 2022-11-13
> **Responses to Reviewer 4ZkJ 1/4**
>
> > 1. In Figure 4, the relationship between $\delta$-hyperbolicity and the train-test gap seems very questionable. In all of the plots, the train-test gap grows while $\delta$-hyperbolicity is shrinking. Additionally, in the fruitbot plot $\delta$-hyperbolicity increases several times with no apparent change in the train-test gap. Furthermore, the scale of the $\delta$-hyperbolicity is the same between representations that learned to successfully solve the task (fruitbot and starpilot) and those that failed to generalize (bigfish and dodgeball), suggesting all of the representations are of a similar level of hyperbolicity. Doesn't that suggest that something else is the cause of the failure to generalize? It definitely does not validate the hypothesis.
>
> While a small generalization gap is still present in fruitbot and starpilot, its magnitude and growth are substantially different from the generalization gap in the bigfish and dodgeball environments. Overall, PPO attains notable improvements and recovers a high test performance in both fruitbot and starpilot (only -12% and -13% as compared to training levels performance) that contrasts its very low test improvements and performance in bigfish and dodgeball (-61% and -63% as compared to training levels performance). This contrast mirrors the observed dichotomy in how hyperbolicity evolves in these two sets of tasks, providing a heuristic measure of whether the latent representations gain or lose hierarchical structure. While we agree that this is most likely not the only factor causing the lack of generalization, we would respectfully argue that these results do appear to be supportive of our hypothesis for the usefulness of encoding hierarchies in RL. **We changed our wording at the end of the fourth paragraph of Section 3.1 to convey less certainty in our analysis**: "*We believe* these results *support* our hypothesis [...]" (from "These results validate our hypothesis [...]")
>
> Regarding the Reviewer's more specific concerns, we would like to note that there are many sources of noise that affect RL optimization and that we tractably estimate $\delta_{rel}$ using the efficient but approximate algorithm from Fournier et al. [1]. Both these factors add noise to the latent representations and hyperbolicity recordings, which are likely the main cause of the small local $\delta_{rel}$ increments observed by the Reviewer in fruitbot. Furthermore, as different RL tasks likely require encoding different amounts of hierarchically-structured information, we believe that relative changes in hyperbolicity should be more informative than its overall scale. To address the Reviewer's concerns, **we have now added Section D.5 to the Appendix, explaining and motivating our assumptions in analyzing the results, as described in this response. In the same new Section,** following the Reviewer's suggestion in Question 8, **we also collected and analyzed the $\delta$-hyperbolicity measurements for our hyperbolic PPO implementation with S-RYM**. We believe also these results are consistent with our analysis: our hyperbolic PPO achieves notably lower values of $\delta_{rel}$, reflecting its higher test performance and lower generalization gap. Furthermore, also in this case, during the iterations where $\delta_{rel}$ increases or attains its higher values, test performance grows significantly slower (see our response to Question 8 for details).
>
>
> > 2. Much of the reasoning in the first part of Section 3.3 seems very speculative. It would be good to adjust the language to better convey that this explanation is a hypothesis.
>
> As suggested, **we edited the language of the first paragraph of Section 3.3 to unequivocally convey that our argument is based on a hypothesis**, e.g.: first sentence "We hypothesize that the..."; last sentence "We believe this is..."
>
>
> > 3. How do standard neural net layers yield Euclidean velocity vectors? This detail seemed to be glossed over.
>
> Given an input $\mathbf{x}$, our architecture first maps this input to a Euclidean latent representation $\mathbf{x}_E$ and then maps this representation to hyperbolic space by using the exponential map from the Poincaré ball's origin, $\mathbf{x}_H=\text{exp}_\mathbf{0}(\mathbf{x}_E)$. The exponential map *treats* its *input* $\mathbf{x}_E$ as the velocity vector of a geodesic (the shortest path) starting at the origin, and *outputs* its end-point in hyperbolic space, $\mathbf{x}_H$. Hence, there is nothing inherently special about $\mathbf{x}_E$, which can be any Euclidean vector outputted by any arbitrary network. **We re-phrased the last paragraph of Section 2.2 to make this clearer**: "we first process the input data $\mathbf{x}$ with standard layers to produce Euclidean vectors $\mathbf{x}_E=f_E(\mathbf{x})$. Then, we obtain our hyperbolic representations by applying the exponential map treating $\mathbf{x}_E$ as a velocity [...]"

---

> > ### Author Response · Authors · 2022-11-13
> > **Responses to Reviewer 4ZkJ 2/4**
> >
> > **Potential weaknesses/questions:**
> >
> >
> > > 4. Some of the figures and most of the plots are too small to read easily. Consider removing the depictions of the environments, and moving the legends to be horizontally laid out underneath the plots. That should allow for larger plots without too much vertical space being used.
> >
> > As suggested by the Reviewer, **we remade our figures by removing the depictions of the environments and increasing the size of the plots**. To further enlarge the figures, we would also be open to making some additional space by moving Figure 8 to the Appendix in case the Reviewer deems it appropriate.
> >
> >
> > > 5. The colour scheme is not accessible to people with colourblindness, and the size of the lines in the legends are too small to easily see their colour.
> >
> > Following the Reviewer's concerns, **we modified all the performance plots to use the color-blind-friendly palette designed by Okabe et al. [2]**.
> >
> >
> > > 6. What are the shaded regions in the plots? Confidence intervals? I wasn't able to find this information.
> >
> > The shaded regions in the plots represent the standard deviation between the different random seeds. We thank the Reviewer for noticing we erroneously omitted to mention this relevant information. **We now added subsection C.4 to the Appendix, providing details of how we record and report the results in our performance curves and tables**.
> >
> >
> > > 7. "our instabilities appear to occur in the gradients from the hyperbolic representations". This was not clear to me from Figure 6. Is there some reason to not use spectral normalization on the hyperbolic representations? Did it hurt performance?
> >
> > We would like to clarify that the gradient with respect to the final layer comes directly from backpropagating through the policy gradient loss, which is dependent on agent performance rather than differences in the models. However, figure 6 (B) shows that it is already after backpropagation through the final 'hyperbolic layer' that its magnitude and variance become orders of magnitude greater as compared to the standard PPO baseline. This difference persists also further backpropagating to the preceding layers as shown by the similar differences observed in the gradients of the convolutional encoder.
> >
> > We also believe there is no direct way of applying spectral normalization (SN) to our hyperbolic layer and that such a practice would most likely not be beneficial for performance. In practice, spectral normalization is implemented by performing power iteration on a layer's weight matrix (e.g., see [3]). However, our final hyperbolic layer does not have an equivalent to this matrix, as its parameters instead represent a hyperplane in hyperbolic space (see Section 2.2). Moreover, by applying SN to all layers we would enforce our models of the value and policy to be 1-Lipschitz, a property likely not reflective of the true optimal policy and value functions. In line with our intuition and other recent RL results (e.g.,[4, 5]), we also found that SN works better when applied to only a subset of the layers for the Euclidean PPO baseline in preliminary experiments.
> >
> > Following the Reviewer's interest, **we extended the last paragraph of Appendix A.3 to discuss the raised points as summarized in this response**. Furthermore, **we added Section E.4 to the Appendix where we evaluate an additional extension of our implementations, applying gradient penalties [6] on top of our hyperbolic PPO with S-RYM and standard PPO**, as an alternative way of enforcing the models to be 1-Lipshitz. Our results further validate our hypothesis and previous results by showing that enforcing either the hyperbolic or Euclidean PPO models to be 1-Lipschitz with this method makes performance collapse across all environments.

---

> > > ### Author Response · Authors · 2022-11-13
> > > **Responses to Reviewer 4ZkJ 3/4**
> > >
> > > **Potential weaknesses/questions:**
> > >
> > >
> > > > 8. Do the representations learned using S-RYM actually exhibit increased $\delta$-hyperbolicity over the base algorithms? I'm very curious about this, but couldn't find this information in the appendices.
> > >
> > > Following the Reviewer's question, **we run additional experiments with our regularized hyperbolic PPO agent, collecting its $\delta$-hyperbolicity**. We analyze these results **in the newly added Section D.5 of the Appendix**. In line with our intuition, using a hyperbolic latent space yields latent representations with significantly lower values of $\delta_{rel}$ implying they possess an increased hierarchical tree-like structure. We observe this trend consistently for all considered tasks and during all stages of the RL training process. The difference in hyperbolicity is particularly evident at initialization, where $\delta_{rel}\approx0.2$ for our hyperbolic PPO while  $\delta_{rel}\approx0.4$ for a standard PPO agent, reflecting how the properties of hyperbolic space facilitate capturing hierarchical relationships even with no training. Interestingly, in the dodgeball environment, the recorded value of $\delta_{rel}$ for our hyperbolic PPO experiences considerable fluctuations in the very initial and later training stages. Analogously to our results in Figure 4, it appears that during the iterations where $\delta_{rel}$ increases or attains its higher values, test performance grows significantly slower. Overall, dodgeball is also the environment with the largest generalization gap relative to the hyperbolic agent's training performance. We summarize these new results and provide a comparison with the original Euclidean PPO results in the Table below:
> > >
> > > | Task\Algorithm | PPO         | PPO        | PPO            | PPO + S-RYM      | PPO + S-RYM      | PPO + S-RYM    |
> > > |-------------------------------|-------------|------------|----------------|------------------|------------------|----------------|
> > > | Metric                        | Train perf. | Test perf. | $\delta_{rel}$ | Train perf.      | Test perf.       | $\delta_{rel}$ |
> > > | bigfish                       | 3.71±1      | 1.46±1     | 0.23±0         | 18.66±6 (+403%) | 13.05±3 (+795%) | 0.17±0 (-26%) |
> > > | dodgeball                     | 5.07±1      | 1.87±1     | 0.24±0         | 7.38±1 (+46%)   | 3.75±1 (+100%)  | 0.22±0 (-11%) |
> > > | fruitbot                      | 30.10±2     | 26.33±2    | 0.24±0         | 30.36±1 (+1%)   | 27.91±1 (+6%)   | 0.20±0 (-16%) |
> > > | starpilot                     | 30.50±5     | 26.57±5    | 0.23±0         | 38.89±2 (+27%)  | 42.15±4 (+59%)  | 0.18±0 (-21%) |
> > >
> > >
> > > > 9. What are some of the shortcomings of the proposed algorithm? I could not find any discussion of limitations.
> > >
> > > Following the Reviewer's question, **we added Section C.5 to the Appendix, where we discuss what we believe are three current limitations of our algorithm**. First, stabilizing hyperbolic representations with S-RYM inherently constrains the expressivity of the Euclidean sub-network encoder model ($f_E$) to be 1-Lipschitz. This loss of expressivity might hinder the network's ability to learn complex representations, preventing our hyperbolic framework from achieving its full potential. Second, the training and evaluation time of our hyperbolic agents are consistently higher than their Euclidean counterparts. Our hyperbolic PPO implementation takes on average 4.27 seconds to collect rollouts and train for three epochs, and takes 0.961 seconds to collect a full episode of experience at the end of training. In contrast, our Euclidean baseline takes 3.69 seconds for training (16% speedup) and 0.854 seconds for evaluation (13% speedup). We find this slowdown is mainly due to the power iteration procedure performed to apply spectral normalization in S-RYM and, to a lesser extent, an overhead when computing and backpropagating through hyperbolic operations. Third, our algorithm utilizes a model of hyperbolic space with fixed negative curvature to build representations of the whole state space. However, as different RL problems might have considerably different structures, we believe that any fixed curvature might not always yield the most appropriate inductive bias. To this end, recent work showed potential benefits in using mixed curvature latent spaces and even learning the curvature parameter for unsupervised tasks [7]. We hope these limitations will be addressed in future work extending our efforts in the study of how differential geometry can be used to empower RL.

---

> > > > ### Author Response · Authors · 2022-11-13
> > > > **Responses to Reviewer 4ZkJ 4/4**
> > > >
> > > > **Minor comments/questions:**
> > > >
> > > >
> > > > > - Bottom of page 2: duplicate "upon".
> > > > > - Bottom of page 3: "of the form in Equation 6" would be clearer.
> > > > > - Figure 6 appears before Figure 5.
> > > >
> > > > **We fixed these minor issues**, removing the second "upon," rewriting the sentence at the end of page 3, and correcting the figure order as suggested.
> > > >
> > > >
> > > > > - What does the dashed line represent in Figure 5?
> > > >
> > > > The dashed line is meant to group the neural network layers in the Impala residual stack, in the Euclidean encoder sub-network ($f_E$), and in the final linear hyperbolic heads ($f_H$). Following the Reviewer's question, **we modified Figure 5 (now Figure 6) by coloring the different groupings in different ways to make the figure clearer to interpret**.
> > > >
> > > >
> > > > > - Start of related work section: "objectives appears has been shown prone to overfitting"
> > > > > - The legend in Figure 8 seems not quite right. The legend indicates dashed lines represent performance on the 200 training levels, but the plots involve a varying number of training levels.
> > > >
> > > > **We corrected these typos**, removing the word "appears" and modifying the label of the dashed lines in the legend of Figure 8 to simply "training levels".
> > > >
> > > >
> > > > **References**
> > > >
> > > > [1] Fournier, Hervé, Anas Ismail, and Antoine Vigneron. "Computing the Gromov hyperbolicity of a discrete metric space." Information Processing Letters 115.6-8 (2015): 576-579.
> > > >
> > > > [2] Okabe, Masataka, and Kei Ito. "Color Universal Design (CUD) - How to make figures and presentations that are friendly to color blind people." University of Tokyo (2002).
> > > >
> > > > [3] Miyato, Takeru, et al. "Spectral Normalization for Generative Adversarial Networks." International Conference on Learning Representations. 2018.
> > > >
> > > > [4] Gogianu, Florin, et al. "Spectral normalisation for deep reinforcement learning: an optimisation perspective." International Conference on Machine Learning. PMLR, 2021.
> > > >
> > > > [5] Bjorck, Nils, Carla P. Gomes, and Kilian Q. Weinberger. "Towards Deeper Deep Reinforcement Learning with Spectral Normalization." Advances in Neural Information Processing Systems 34 (2021): 8242-8255.
> > > >
> > > > [6] Gulrajani, Ishaan, et al. "Improved training of wasserstein gans." Advances in neural information processing systems 30 (2017).
> > > >
> > > > [7] Skopek, Ondrej, Octavian-Eugen Ganea, and Gary Bécigneul. "Mixed-curvature Variational Autoencoders." International Conference on Learning Representations. 2019.

---

### Official Review · Reviewer_7pBh · 2022-11-03

**Confidence:** 4
**Correctness:** 3
**Technical Novelty And Significance:** 3
**Empirical Novelty And Significance:** 3
**Recommendation:** 10

**Clarity, Quality, Novelty And Reproducibility:**

This paper is nicely written for motivation and stating the results. However, the reviewer cannot grasp the implementation details of this paper which hinders the reviewer’s evaluation of the novelties of this paper. The reviewer believes the empirical results of this work are reproducible (although the reviewer did not run the code).

**Strength And Weaknesses:**

Strength:
1. As an empirical paper, it demonstrates superior performance on many benchmarks (Atari and Procgen).
2. It illustrates the connection between the hierarchical structure in hyperbolic representation and reinforcement learning

Weakness/Questions:
1. To the best of the reviewer’s understanding, the implementation details of this paper are not clearly stated:
(a) If the reviewer understands the implementation clearly, it seems that the implementation adopts the spectral normalization to the hydra package (https://github.com/facebookresearch/hydra)? The reviewer would appreciate it if the author can clarify the novelty in the implementation for better readability! (b) Regardless of whether the novelty in implementation is purely “spectral normalize+hydra”, the author should at least cite the Hydra Package if they are using it as an implementation backbone. (c) The reviewer guesses the last paragraph of section 2.2 “In line with recent use of hyperbolic geometry in supervised …” characterizes the main contents of the main implementation details of the hyperbolic embedding, but perhaps the authors can elaborate more on this part so that the main contribution of this work is much better than  “spectral normalization + some hyperbolic representation learning method other people proposed”.
2. The motivation from $\delta$-hyperbolicity to the pursuit of hyperbolic representation is very inspiring. The reviewer is wondering whether the authors can reproduce Figure 4 using PPO + S-RYM. If the authors can demonstrate the $\delta$-hyperbolicity decreases with PPO + S-RYM, it would also improve the results of this work.
3. How does the implementation of the hyperbolic embeddings different from other online packages (e.g., https://github.com/nalexai/hyperlib)?

**Summary Of The Paper:**

This paper proposes spectrally-regularized hyperbolic mappings (S-RYM) to learn a hyperbolic representation for deep RL, by applying a spectral normalization for learning the hyperbolic representation.

**Summary Of The Review:**

In summary, since the empirical results of this work beat the SOTA, the reviewer believes it definitely reaches the bar for acceptance. However, the reviewer believes there is a large room for the writing part (for clarifying the methods and addressing how the proposed method is different from SN + Hydra) so that other readers to better appreciate the merits of this work.

---

> ### Author Response · Authors · 2022-11-13
> **Responses to Reviewer 7pBh 1/2**
>
> **Weaknesses/Questions**
>
>
> > 1. To the best of the reviewer’s understanding, the implementation details of this paper are not clearly stated:
>
> > (a) If the reviewer understands the implementation clearly, it seems that the implementation adopts the spectral normalization to the hydra package (https://github.com/facebookresearch/hydra)? The reviewer would appreciate it if the author can clarify the novelty in the implementation for better readability!
>
> The proposed implementation (S-RYM) stabilizes learning with hyperbolic representations using two main methods. First, as noted by the Reviewer, we apply spectral normalization to all layers *before* the final latent representations. Second, we also propose to scale these latent representations to account for their dimensionality, effectively regulating their expected magnitude. We finally apply an exponential map and compute the policy and value with a learned linear transformation in hyperbolic space, as depicted in Figure 5. Following the Reviewer's suggestion, **we modified the third paragraph of Section 3.3 to make the novelties of S-RYM clearer**: "Our implementation differs from [spectral normalization's] usual application for GANs in two main ways. First, we apply SN to all layers in the Euclidean sub-network ($f_E$) [...]. Second, we propose to scale the latent representations to account for their dimensionality [...]."
>
>
> > (b) Regardless of whether the novelty in implementation is purely “spectral normalize+hydra”, the author should at least cite the Hydra Package if they are using it as an implementation backbone.
>
> The Hydra package [1] offers a configuration framework to run Python code by composing ".yaml" files in a modular and practical way but does not provide any scientific functionality related to machine learning or hyperbolic geometry. We use the Hydra library only to facilitate storing hyper-parameters and quickly specifying the different algorithm variations for our experiments. As suggested by the Reviewer, **we now added to the second paragraph of Appendix C a reference to the Hydra package together with a description of its utilization in the shared code**.
>
>
> > (c) The reviewer guesses the last paragraph of section 2.2 “In line with recent use of hyperbolic geometry in supervised …” characterizes the main contents of the main implementation details of the hyperbolic embedding, but perhaps the authors can elaborate more on this part so that the main contribution of this work is much better than “spectral normalization + some hyperbolic representation learning method other people proposed”.
>
> Section 2.2 provides the background for our implementation of a linear hyperbolic layer, using the signed distance to a parameterized hyperplane in hyperbolic space, following the work of Ganea et al. [2]. To the best of our knowledge, our work is the first to consider applying this methodology for reinforcement learning, parameterizing the final layer of the policy and value models. The inspiration for this implementation comes from applications of similar methodologies in some prior work focused on using hyperbolic geometry for other areas of machine learning, as referenced in the last paragraph of the section. We would like to highlight that together with SN, S-RYM also entails a particular dimensionality-based rescaling we designed for the effective stabilization of hyperbolic embeddings, described in Section 3.3. We show that *both* SN and rescaling are essential to obtain good performance in our ablation study in Section E.1 of the Appendix. To the best of our knowledge, no prior work ever considered stabilizing hyperbolic representation learning with similar regularization techniques. We provide empirical results in Appendix B.2 showing that the benefits of S-RYM extend beyond the non-stationary RL setting, also improving performance in image classification problems in two datasets. Following the Reviewer's comment, **we rewrote the last paragraph of Section 2.2 to better characterize our contribution also by adding an earlier mention to our new stabilization technique**, e.g.: "[...] We extend this model with a new stabilization (Sec 3.3) with observed benefits beyond RL (App. B)"

---

> > ### Author Response · Authors · 2022-11-13
> > **Responses to Reviewer 7pBh 2/2**
> >
> > **Weaknesses/Questions**
> >
> >
> > > 2. The motivation from $\delta$-hyperbolicity to the pursuit of hyperbolic representation is very inspiring. The reviewer is wondering whether the authors can reproduce Figure 4 using PPO + S-RYM. If the authors can demonstrate the $\delta$-hyperbolicity decreases with PPO + S-RYM, it would also improve the results of this work.
> >
> > Following the Reviewer's suggestion, **we run additional experiments with our regularized hyperbolic PPO agent, collecting its $\delta$-hyperbolicity**. We analyze these results **in the newly added Section D.5 of the Appendix**. In line with our intuition, using a hyperbolic latent space yields latent representations with significantly lower values of $\delta_{rel}$, implying they possess an increased hierarchical tree-like structure. We observe this trend consistently for all considered tasks and during all stages of the RL training process. The difference in hyperbolicity is particularly evident at initialization, where $\delta_{rel}\approx0.2$ for our hyperbolic PPO while  $\delta_{rel}\approx0.4$ for a standard PPO agent, reflecting how the properties of hyperbolic space facilitate capturing hierarchical relationships even with no training. Interestingly, in the dodgeball environment, the recorded value of $\delta_{rel}$ for our hyperbolic PPO experiences considerable fluctuations in the very initial and later training stages. Analogously to our results in Figure 4, it appears that during the iterations where $\delta_{rel}$ increases or attains its higher values, test performance grows significantly slower. Overall, dodgeball is also the environment with the largest generalization gap relative to the hyperbolic agent's training performance. We summarize these new results and provide a comparison with the original Euclidean PPO results in the Table below:
> >
> > | Task\Algorithm | PPO         | PPO        | PPO            | PPO + S-RYM      | PPO + S-RYM      | PPO + S-RYM    |
> > |-------------------------------|-------------|------------|----------------|------------------|------------------|----------------|
> > | Metric                        | Train perf. | Test perf. | $\delta_{rel}$ | Train perf.      | Test perf.       | $\delta_{rel}$ |
> > | bigfish                       | 3.71±1      | 1.46±1     | 0.23±0         | 18.66±6 (+403%) | 13.05±3 (+795%) | 0.17±0 (-26%) |
> > | dodgeball                     | 5.07±1      | 1.87±1     | 0.24±0         | 7.38±1 (+46%)   | 3.75±1 (+100%)  | 0.22±0 (-11%) |
> > | fruitbot                      | 30.10±2     | 26.33±2    | 0.24±0         | 30.36±1 (+1%)   | 27.91±1 (+6%)   | 0.20±0 (-16%) |
> > | starpilot                     | 30.50±5     | 26.57±5    | 0.23±0         | 38.89±2 (+27%)  | 42.15±4 (+59%)  | 0.18±0 (-21%) |
> >
> > >3. How does the implementation of the hyperbolic embeddings different from other online packages (e.g., https://github.com/nalexai/hyperlib)?
> >
> > We implement the proposed dimensionality-based rescaling followed by the exponential map and the linear hyperbolic layer as a single Pytorch module (the PoincarePlaneDistance class in the code). This allows us to easily extend existing neural network architectures to make use of hyperbolic geometry by swapping the final layer with our module. As mentioned in Appendix C, we employ the GeoOpt library [3] to efficiently optimize our network's parameters directly in hyperbolic space. The HyperLib library [4] referenced by the Reviewer implements a hyperbolic Tensorflow layer by storing its parameters in Euclidean space and applying an additional exponential map during every evaluation. These additional operations could lead to inefficiencies and computational approximations that our implementation is able to circumvent. Following the Reviewer's remark, **we added a new paragraph at the beginning of Appendix C providing these details**. While we already shared the code to reproduce our main experiments, we plan to open-source a fully-documented version of our implementation after the review period to facilitate future applications of hyperbolic representations for RL.
> >
> >
> > **References**
> >
> > [1] Yadan, Omry. "Hydra-a framework for elegantly configuring complex applications." Github (2019).
> >
> > [2] Ganea, Octavian, Gary Bécigneul, and Thomas Hofmann. "Hyperbolic neural networks." Advances in neural information processing systems 31 (2018).
> >
> > [3] Kochurov, Max, Rasul Karimov, and Serge Kozlukov. "Geoopt: Riemannian optimization in pytorch." arXiv preprint arXiv:2005.02819 (2020).
> >
> > [4] Francis, Nathan, and Joseph, Alex "HyperLib: Deep learning in the Hyperbolic space." Github (2021).

---

> > > ### Comment · Reviewer_7pBh · 2022-11-14
> > > **All concerns are addressed and rating has been raised accordingly**
> > >
> > > Dear Authors,
> > >
> > > Thank you very much for the updates. Now the reviewer has understood the contributions much better -- indeed it s a solid contribution to the community hence the rating has been raised accordingly.
> > >
> > > Also, a small suggestion -- perhaps reference [3] (Kochurov, Max, Rasul Karimov, and Serge Kozlukov. "Geoopt: Riemannian optimization in pytorch." arXiv preprint arXiv:2005.02819 (2020)) should also be cited, as you have adopted their implementation.

---

> > > > ### Author Response · Authors · 2022-11-14
> > > > **Further Response to Reviewer 7pBh**
> > > >
> > > > We thank the Reviewer for all their relevant and constructive criticism to improve the quality of our work.  In the latest revision, we fixed the reference to the Geoopt library [3] as suggested.

---

### Author Response · Authors · 2022-11-13
**General Response**

We thank the Reviewers for the time dedicated to our work, and for providing insightful and actionable feedback. Furthermore, we are very glad that all Reviewers seemed to have a positive impression of the novelty and quality of our work.  We address each of the raised points in our individual responses below, where we **emphasized** the resulting additions and modifications made to our paper. Here, we summarize the main changes:

- (**7pBH**) We modified Section 3.3 to make the novelty of the proposed stabilization method (S-RYM) clearer.
- (**7pBH**) In Appendix C, we now detail the characteristics of our implementation, including a reference and description of our use of the Hydra package.
- (**7pBH, 4ZkJ**) We rewrote Section 2.2 to better characterize our contribution and clarify how we use the Euclidean representations as velocity vectors.
- (**7pBH, 4ZkJ**) In the newly added Section D.5 of the Appendix, we provide new results and analysis for the $\delta$-hyperbolicity of our hyperbolic PPO agent. We also provide additional discussion of our interpretation of the $\delta$-hyperbolicity results.
- (**4ZkJ**) We edited the language of Sections 3.1 and 3.3, to better convey the provided analysis and claims are based on our interpretation.
- (**4ZkJ**) We remade all the performance plots, increasing the size of the figures and adopting a color-blind-friendly color scheme.
- (**4ZkJ**) In the newly added Section C.4 of the Appendix, we provide details on how we record and report our results.
- (**4ZkJ**) In Appendix A.3, we now discuss our implementation choices with respect to keeping the final layer of S-RYM unregularized. Furthermore, in the newly added Section E.4 of the Appendix, we empirically evaluate the effects of further regularization of our agents.
- (**4ZkJ, 56hh**) In the newly added Section C.5 to the Appendix, we discuss three current limitations of our algorithm, including how different state relationships in RL might influence the optimal representation space.
- (**56hh**) We edited Section 3.3 to better decribe our Euclidean PPO + S-RYM baseline.
- (**56hh**) We added qualitative visualizations of the learned 2-dimensional representations of our hyperbolic agent to Appendix D.4.

We hope to have addressed all the raised concerns and would be happy to respond to further questions and suggestions.

---

### Decision · Program_Chairs · 2023-01-20

**Decision:**

Accept: notable-top-25%

**Justification For Why Not Higher Score:**

Even though two reviewers gave a rating of 10, I think the reviewers' reviews still indicate various (at least minor) issues with the paper. The AC also thinks it would be useful to conduct experiments on other types of RL environments. This is why the AC thinks spotlight might be the best category for this paper.

**Justification For Why Not Lower Score:**

Strength:

"As an empirical paper, it demonstrates superior performance on many benchmarks (Atari and Procgen).
It illustrates the connection between the hierarchical structure in hyperbolic representation and reinforcement learning"

---"The paper is clearly written and easy to follow."
---"The idea of learning latent representations using hyperbolic space for RL is novel, and successfully "
---"adapting hyperbolic deep learning methods to RL is non-trivial."
---"The empirical performance improvements on Procgen and Atari are significant."



**Metareview: Summary, Strengths And Weaknesses:**

Strength:

"As an empirical paper, it demonstrates superior performance on many benchmarks (Atari and Procgen).
It illustrates the connection between the hierarchical structure in hyperbolic representation and reinforcement learning"

---"The paper is clearly written and easy to follow."
---"The idea of learning latent representations using hyperbolic space for RL is novel, and successfully "
---"adapting hyperbolic deep learning methods to RL is non-trivial."
---"The empirical performance improvements on Procgen and Atari are significant."



**Note From Pc:**

if the above contains the word "oral" or "spotlight" please see: "oral" presentation means -> notable-top-5% and "spotlight" means -> notable-top-25%. As stated in our emails, we are disassociating presentation type from AC recommendations